# Finding Harmony in Chemical Data: Hierarchical and Balanced multimodal Fusion for Reaction Yield Prediction

## Abstract

Multimodal yield prediction aims to integrate heterogeneous molecular descriptors across distinct data modalities to predict the conversion efficiency of chemical reactions. However, existing approaches often face limitations in effectively utilizing multimodal information, primarily due to inadequate consideration of both hierarchical relationships and imbalanced contributions across modalities during the fusion process. To address these challenges, we propose a **H**ier**ar**chical and balanced **m**ulti-m**o**dal fusion framework for reactio**n** **y**ield prediction, termed **Harmony**. Specifically, to enhance multimodal information utilization, we design a hierarchical fusion architecture comprising three modality encoders and two feature fusion modules for different levels of granularity. Furthermore, we introduce a novel contribution assessment mechanism that quantitatively evaluates modality-specific impacts, coupled with a prefer-balancing optimization objective. Extensive experimental evaluations demonstrate that Harmony not only consistently outperforms existing methods but also exhibits robust out-of-sample (OOS) generalization. Specifically, it achieves a **22% improvement** in the $R^2$ metric over the strongest baseline on the most challenging Amide Coupling Reaction dataset. Our code can be found at `https://anonymous.4open.science/r/F6BB`.

## 1 Introduction

Predicting chemical reaction yields is a central challenge in AI-driven synthesis planning (Johansson et al., 2019). Early models relied on a single molecular modality, such as SMILES strings (Chuang & Keiser, 2018; Schwaller et al., 2020) or 2D molecular graphs (Kwon et al., 2022), but such unimodal representations are intrinsically limited, as no single view can capture the full spectrum of structural, topological, and electronic factors governing reaction outcomes. To overcome this limitation, recent research has advanced toward multimodal fusion, integrating complementary representations including sequential SMILES and fingerprint-based descriptors (Weininger, 1988; Rogers & Hahn, 2010). This paradigm better reflects the heterogeneity of chemical information and has yielded marked improvements in prediction accuracy (Chen et al., 2024; Shi et al., 2024).

A common yet often overlooked challenge in multimodal reaction modeling is the standard "flat fusion" approach. Current architectures typically treat all input modalities as equals, mixing features that represent fundamentally different levels of chemical abstraction. This approach creates a conflict by processing low-level, molecule-specific details like the atomic structure from SMILES strings and 2D graphs, alongside high-level, reaction-wide patterns found in reaction fingerprints. Such a strategy ignores the inherent hierarchical nature of chemical information, leading to feature interference where the crucial, abstract signals are obscured by low-level noise, severely constraining the model's ability to learn robust representations.

This architectural flaw causes a critical issue we term representational overshadowing, a domain-specific manifestation of the "modality preference" problem (Huang et al., 2022). In this context, the model's predictions become dominated by abstract representations (e.g., reaction fingerprints), effectively silencing signals from foundational structural data and reducing the system to a unimodal model (Yang et al., 2024; Wei et al., 2024). As shown in prior work UAM (Chen et al., 2024), removing the high-level fingerprint features leads to a sharp 19.1% drop in performance, whereas removing other modalities causes only a minor decline of 0.4%–0.8% (Figure 1a). However, such

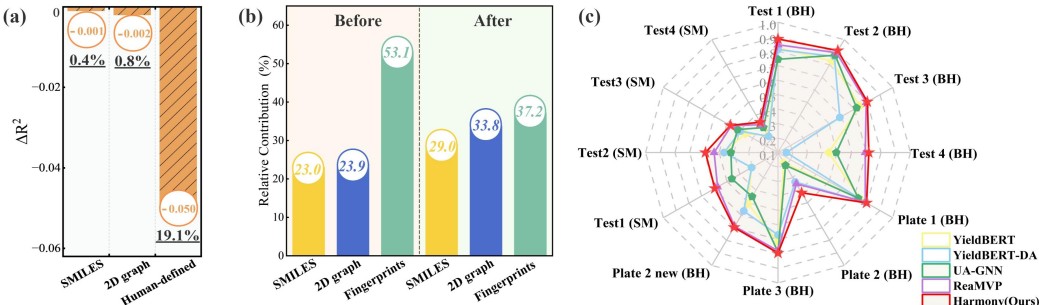

Figure 1: **(a):** The decrease in $R^2$ occurs upon the removal of a single modality from the UAM framework. **(b):** The contribution of each modality estimated using our method before and after balancing modality contributions. **(c):** Ligand-Based Out-of-Sample Results for the Buchwald-Hartwig (BH) and Suzuki-Miyaura (SM) Datasets.

analyses merely reveal the symptom, as conventional ablation studies fail to quantify the dynamic contributions within the fusion process. This highlights a foundational flaw in multi-scale chemical modeling, demanding a shift from simple to intelligent, hierarchy-aware combinations.

To pioneer this shift, we propose a **hi**erar**ch**ical and balanced **m**ulti-m**o**dal fusion framework for reactio**n y**ield prediction, termed **Harmony**. Our framework tackles this issue through a staged process that mirrors the natural hierarchy of chemical information. It first combines molecular-level data (SMILES and 2D graphs) into a complete molecular representation. Then it moves to the reaction level, merging this unified view with reaction fingerprints. This granularity-aware design enforces a structured information flow, fundamentally preventing the representational overshadowing endemic to flat architectures.

Beyond architectural innovation, we establish an integrated diagnostic and optimization framework to validate and enforce balanced fusion. Using causal inference, we first quantify the contributions of different representations in conventional models and reveal a pronounced imbalance: abstract fingerprints dominate with 53.1% of the influence (Figure 1b, left). Building on this causal insight, we design a preference-balancing optimization objective that directly counteracts such dominance. In combination with our hierarchy-aware architecture, this leads to a near-uniform contribution distribution ($\approx 33.3\%$ each), ensuring that every level of chemical information is not only preserved but fully engaged. Together, these advances represent a significant step toward truly multi-scale chemical intelligence (Figure 1b, right).

To our knowledge, Harmony represents one of the first systematic attempts in AI for Chemistry to integrate hierarchical fusion and quantitative modality assessment in reaction yield prediction. This principled approach directly addresses the critical issue of representational overshadowing, leading to superior generalization capabilities. This is particularly evident in its out-of-sample performance, where Harmony significantly outperforms existing models when confronted with molecules entirely absent from the training data (Figure 1c). This robust generalization translates directly into state-of-the-art (SOTA) performance across three of the most widely-used benchmark datasets, establishing a new performance ceiling for reaction yield prediction.

Our main contributions are summarized as follows:

- **(Hierarchy)** We introduce Harmony, a hierarchical and balanced multimodal fusion framework for reaction yield prediction that mirrors the natural hierarchy of chemical information, integrating molecular-level representations with reaction fingerprints to ensure structured information flow and prevent representational dominance.

- **(Balance)** We develop a causal-inference–guided diagnostic and preference-balancing optimization framework that, together with our hierarchy-aware architecture, equalizes modality contributions and ensures real engagement of all chemical information levels, advancing toward truly multi-scale chemical intelligence.

- **(Validation)** Extensive experiments demonstrate that Harmony consistently achieves state-of-the-art performance on three benchmark datasets and exhibits strong generalization to out-of-sample data, highlighted by a remarkable 22% improvement in the $R^2$ metric on the most challenging dataset compared to the second-best method.

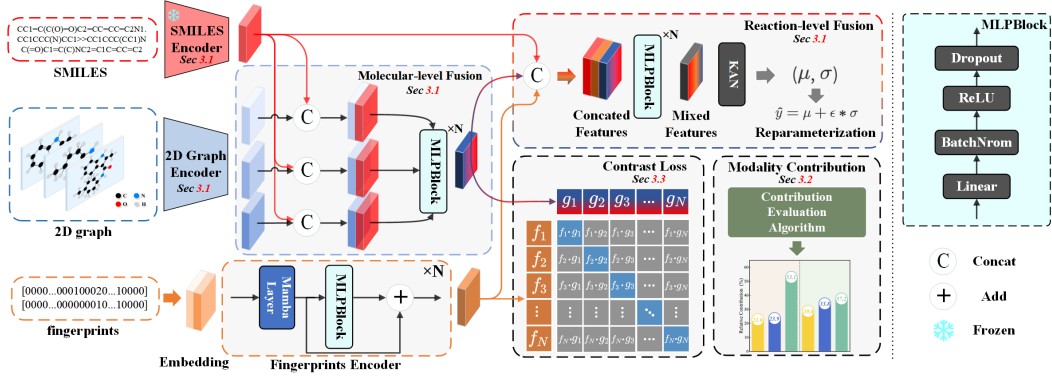

Figure 2: Harmony integrates modality-specific feature encoders and multi-tiered fusion modules across both molecular and reaction levels, complemented by a modality contribution evaluation mechanism and alignment at the molecular-reaction level.

## 2 RELATED WORK

**Multimodal Learning in Chemistry and its Pitfalls** Multimodal learning is increasingly vital for advancing chemistry-related tasks. In reaction yield prediction, the field has progressed from unimodal models (Schwaller et al., 2021) to sophisticated methods integrating SMILES, 2D graphs, and 3D conformers (Shi et al., 2024; Chen et al., 2024). This trend extends to other areas, such as molecular characterization (Luo et al., 2023) and text-guided generation (Guo et al., 2024). Despite their success, a common architectural flaw pervades these models: they employ a "flat" fusion strategy that treats all modalities as equals. This approach ignores the vast differences in real chemical granularity between inputs (e.g., molecular-level graphs vs. reaction-level fingerprints), often causing models to rely on a single dominant modality and degenerate into de-facto unimodal systems.

**Modality Imbalance** This phenomenon, often termed modality preference or imbalance, is a recognized challenge in the broader multimodal learning community, observed in tasks like visual question answering (Huang et al., 2022). Recent efforts have focused on quantifying modality contributions (Wei et al., 2024) or redesigning fusion mechanisms to enhance robustness against this bias (Yang et al., 2024). However, these general-purpose solutions are not tailored to the unique, inherent informational hierarchy of chemical data. To our knowledge, the problem of modality imbalance arising from the fusion of hierarchically distinct chemical representations has not been systematically addressed, representing a critical gap we aim to fill.

## 3 METHOD

In this study, we introduce Harmony, a novel hierarchical multimodal fusion model designed for reaction yield prediction, with its architecture illustrated in Figure 2. We begin by introducing the design of the hierarchical fusion framework, encompassing modality encoders, molecular-level and reaction-level fusion modules. Subsequently, we discuss the modality contribution assessment method using causal graphs and counterfactual reasoning. Finally, we outline the optimization objectives for Harmony, including a prefer-balancing objective to mitigate modality preference.

### 3.1 ENCODING AND FUSING DIVERSE MOLECULAR MODALITIES

In this subsection, we present the three-level hierarchical multimodal fusion framework. It starts with unique encoders for SMILES, 2D graphs, and fingerprints to extract modality-specific information. The intermediate layer merges the SMILES of chemical reaction and 2D graphs of molecular for a blend of global and detailed molecular information, while the top layer further integrates the fingerprints modality, which carries compressed reaction information.

**SMILES Encoder** SMILES are string sequences that encode molecular structures and reactions. With the rapid progress of NLP and Transformer architectures (Vaswani, 2017; Qin et al., 2024), large-scale pre-trained SMILES encoders have become increasingly powerful. Training such models from scratch on limited yield data risks overfitting to sequence grammar rather than capturing the

underlying chemical reactivity. To address this, we adopt the pre-trained ChemBERTa-2 (Ahmad et al., 2022) as our SMILES encoder, $\mathrm{Enc}^{\mathrm{s}}(\cdot)$, enabling extraction of features $\boldsymbol{h}^{\mathrm{s}}i$ from reaction SMILES $\boldsymbol{x}^{\mathrm{s}}i$ while leveraging its rich chemical prior knowledge (details in Appendix P). This integration grounds our framework in both modern AI advances and domain-specific chemical insights. Let the notation $\boldsymbol{x}_i$ denote a sample among the total $N$ samples, the encoding process follows as:

$$\boldsymbol{h}_i^{\mathrm{s}} = \mathrm{Enc}^{\mathrm{s}}(\boldsymbol{x}_i^{\mathrm{s}}), \, i \in \{1, \dots, N\} \,. \tag{1}$$

**2D Graph Encoder**  In a molecular 2D graph, nodes and edges symbolize atoms and binds respectively. The graph encoder, denoted as $\mathrm{Enc}^{\mathrm{g}}(\cdot)$, extracts features from a 2D graph by aggregating local node and edge information, and employs graph-level average pooling for global insights. For each reaction sample $\boldsymbol{x}_i$, let $T_i$ denote the number of distinct molecular species involved. The notation $\boldsymbol{x}_{i,j}^{\mathrm{g}}$ denotes the 2D graph of the $j$-th molecule, and its encoded features, denoted by $\boldsymbol{h}_{i,j}^{\mathrm{g}}$, can be derived as follows:

$$\boldsymbol{h}_{i,j}^{\mathrm{g}} = \mathrm{Enc}^{\mathrm{g}}(\boldsymbol{x}_{i,j}^{\mathrm{g}}), \, i \in \{1, \dots, N\}, \, j \in \{1, \dots, T_i\} \,. \tag{2}$$

**Molecular-level Fusion Module**  Encoding each 2D graph directly fails to consider the context of a reaction, due to the same molecule has the same encoding results even in different reactions. To address this limitation, we develop a molecular fusion module where reaction-contextualized SMILES features $\boldsymbol{h}_i^{\mathrm{s}}$ are integrated with structural information through a multi-layer perceptron (MLP) denoted as $\mathrm{MLPBlock}(\cdot)$ comprising linear transformation, batch normalization, ReLU activation, and dropout. The result of the fusion between molecular structural information and global information, denoted as $\widetilde{\boldsymbol{h}}_{i,j}^{\mathrm{g}}$, can be obtained as follows:

$$\widetilde{\boldsymbol{h}}_{i,j}^{\mathrm{g}} = \mathrm{MLPBlock}(\boldsymbol{h}_{i,j}^{\mathrm{g}} \oplus \boldsymbol{h}_i^{\mathrm{s}}), \, i \in \{1, \dots, N\}, \, j \in \{1, \dots, T_i\}, \tag{3}$$

where $\boldsymbol{a} \oplus \boldsymbol{b}$ denotes the concatenation of vector $\boldsymbol{a}$ and $\boldsymbol{b}$. We aggregate molecular-level features through summation, transforming variable-length molecular graph feature sequences into a fixed-length vector $\boldsymbol{h}_i^{\mathrm{g}}$ and ensuring representation invariance to reactant order:

$$\boldsymbol{h}_i^{\mathrm{g}} = \sum_{j=1}^{T} \widetilde{\boldsymbol{h}}_{i,j}^{\mathrm{g}} \,, \, i \in \{1, \dots, N\} \,. \tag{4}$$

**Fingerprints Encoder**  The extended-connectivity fingerprints are represented by a fixed-length sequence of bits, with each bit referring to structures or properties that a molecule possesses (Yang et al., 2022). To encode this modality, we apply embedding function $\mathcal{E}(\cdot)$ to turn each bit into a vector and use a Mamba-based (Gu & Dao, 2023) backbone $\mathrm{Enc}^{\mathrm{f}}(\cdot)$ to extract features. Mamba is a cutting-edge selective structured state space model that can significantly improve the processing speed for long sequences by ensuring linear scalability with the length of the sequence (Qu et al., 2024). Specifically, we compute reactant fingerprints $\boldsymbol{x}_i^{\mathrm{f,r}}$ and product fingerprints $\boldsymbol{x}_i^{\mathrm{f,p}}$ through element-wise summation of individual molecular fingerprints. The concatenated vector $[\boldsymbol{x}_i^{\mathrm{f,r}}, \boldsymbol{x}_i^{\mathrm{f,p}}]$ is then processed by a Mamba-based encoder to capture both global reaction characteristics and dynamic transformation patterns. A detailed analysis is provided in Appendix F.6. The encoding results $\boldsymbol{h}_i^{\mathrm{f}}$ can be defined as:

$$\boldsymbol{h}_i^{\mathrm{f}} = \mathrm{Enc}^{\mathrm{f}}(\mathcal{E}([\boldsymbol{x}_i^{\mathrm{f,r}}, \boldsymbol{x}_i^{\mathrm{f,p}}]), \, i \in \{1, \dots, N\} \,. \tag{5}$$

**Reaction-level Fusion and Prediction**  After extracting features from each modality and integrating molecular-level features, we concatenate these multimodal features and employ a late fusion module (Snoek et al., 2005) for yield prediction. We replace the traditional MLP with a Kolmogorov-Arnold network (KAN) layer(Liu et al., 2024) in the final layer of the late fusion module. Unlike MLPs that apply activation functions on neuron nodes, KAN implements learnable activation functions between weight connections. Specifically, it parameterizes these weights using B-spline basis functions, enabling more flexible feature transformations. By leveraging its powerful non-linear mapping capabilities, it can make better use of the mixed multimodal features than a linear layer. The late fusion module can be defined as the composite function $\mathcal{F}(\cdot) = \mathrm{KAN}(\mathrm{MLPBlock}(\cdot))$.

It is widely acknowledged that the yield of a chemical reaction varies according to the molecule's configuration and conformation, as well as the reaction conditions. Follow the previous works

(Kwon et al., 2022; Chen et al., 2024), we adopt a strategy to predict a range for yield instead of a single, definitive value. For any given reaction sample $\boldsymbol{x}_i$, our model will provide the mean and variance of the yield corresponding to the sample, denoted as $\mu(\boldsymbol{x}_i)$ and $\sigma(\boldsymbol{x}_i)$, respectively. The final multimodal late fusion and prediction process and be formalized as:

$$(\mu(\boldsymbol{x}_i),\, \sigma(\boldsymbol{x}_i)) = \mathcal{F}(\boldsymbol{h}_i^{\mathrm{s}} \oplus \boldsymbol{h}_i^{\mathrm{g}} \oplus \boldsymbol{h}_i^{\mathrm{f}}),\, i \in \{1, \ldots, N\}. \tag{6}$$

For a more general representation, we define $\mathcal{M}$ as the set of all modalities, and $m$ as one of these modalities. The relationship between them and the SMILES, 2D graph, and fingerprints discussed in this subsection can be expressed as $\mathcal{M} = \{m | m \in \{\mathrm{s}, \mathrm{g}, \mathrm{f}\}\}$. The features from modality $m$ are denoted as $\boldsymbol{h}_i^m$ and Equation equation 6 can be redefined as:

$$(\mu(\boldsymbol{x}_i),\, \sigma(\boldsymbol{x}_i)) = \mathcal{F}\left( \bigoplus_{m \in \mathcal{M}} \boldsymbol{h}_i^m \right),\, i \in \{1, \ldots, N\}. \tag{7}$$

The predicted yield $\hat{y}_i$ is then derived using a reparameterization trick (Kingma, 2013): $\hat{y}_i = \mu(\boldsymbol{x}_i) + \epsilon * \sigma(\boldsymbol{x}_i)$, where $\epsilon$ is sampled from a standard normal distribution.

## 3.2 MODALITY CONTRIBUTION EVALUATION

Each modality reflects distinct aspects of molecular information, and multimodal learning can fully utilize this complementary information (Jiao et al., 2024). However, during the actual fusion process, we find that the model tends to rely on specific modalities, neglecting or even suppressing heterogeneous information in other modalities. This results in the loss of advantages gained from integrating information from multiple sources, causing the model to degenerate into a unimodal one. To figure out the effects of different modalities, in this subsection, we propose an algorithm to evaluate the contributions of each modality.

**Contribution of the Modality Subset** $\mathcal{C}$ Supposing we have $n$ modalities, the unordered set of all modalities is denoted as $\mathcal{M} = \{m_1, m_2, \ldots, m_n\}$. Let $\mathcal{C}$ as a subset of all modalities that we wish to evaluate, thus we have $\mathcal{C} \subset \mathcal{M}$. To simplify the problem, let $\boldsymbol{x}$ denote any given sample, ignoring its subscript. Furthermore, the late fusion module $\mathcal{F}(\cdot)$ directly produces the yield prediction outcome $\hat{y}$. Based on Equation equation 7, we can derive the formula for predicting outcomes using the modality subset $\mathcal{C}$ as follows:

$$\hat{y}^{\mathcal{C}} = \mathcal{F}\left( \bigoplus_{m \in \mathcal{M}} \mathcal{T}(\boldsymbol{x}^m; \mathcal{C}) \right), \tag{8}$$

where $\hat{y}^{\mathcal{C}}$ is the yield prediction results using modality subset $\mathcal{C}$, and $\boldsymbol{x}^m$ is the value of modality $m$ for sample $\boldsymbol{x}$. $\mathcal{T}(\boldsymbol{x}^m; \mathcal{C})$ is a mapping function, whose parameters represent the data of modality $m$ of sample $\boldsymbol{x}$ and the modality subset $\mathcal{C}$. It is defined as follows:

$$\mathcal{T}(\boldsymbol{x}^m; \mathcal{C}) = \begin{cases} \Phi^m(\boldsymbol{x}^m) & m \in \mathcal{C}, \\ \boldsymbol{0} & m \notin \mathcal{C}, \end{cases} \tag{9}$$

where the function $\Phi^m(\cdot)$ represents the feature extraction network for modality $m$. Equation equation 9 indicates that for any modality $m \in \mathcal{M}$, if $m$ also belongs to $\mathcal{C}$, then $\boldsymbol{x}^m$ will be processed by the corresponding feature extractor. Otherwise, it will be mapped to a zero vector that has the same shape as $\boldsymbol{x}^m$.

Given the predicted yield $\hat{y}^{\mathcal{C}}$ and the real yield $y$, we define the contribution of the modality subset $\mathcal{C}$ as follows:

$$\mathcal{B}(\mathcal{C}) = \begin{cases} |\mathcal{C}| \cdot \xi & \text{if } |\hat{y}^{\mathcal{C}} - y| \leq \varepsilon, \\ 0 & \text{otherwise}. \end{cases} \tag{10}$$

Our formulation introduces a tolerance threshold $\varepsilon$ and a smoothing factor $\xi$. Intuitively, $\varepsilon$ specifies the acceptable error margin under which a modality subset is considered to have "explained" the outcome, while $\xi$ ensures continuous attribution instead of binary inclusion/exclusion. Notably, the parameter $\varepsilon$ is used for observational purposes only and does not participate in model training. Although heuristic, this design is motivated by the need for stable estimation in regression tasks, where exact Shapley-style computation is intractable. The symbol $\xi$ signifies the smoothing factor to make the contribution smoother and continuous, which is defined as follows:

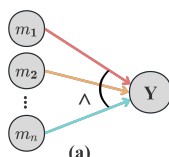 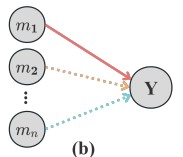 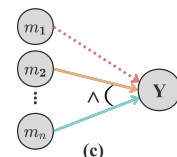

Figure 3: **Different reasoning methods and causal diagrams are explored.** Let $\{m_1, ..., m_n\}$ denote $n$ modalities and $Y$ the prediction outcome. Solid/dashed arrows represent presence/absence of causal effects, and $\wedge$ indicates joint causation. (a) Full-modality prediction using all $m_i$; (b) Forward reasoning considering only $m_1$'s effect; (c) Counterfactual analysis excluding $m_1$.

$$\xi = \min\left(1,\, 2 \cdot \log \frac{\varepsilon}{|\hat{y}^{\mathcal{C}} - y| + \delta}\right), \tag{11}$$

where $\delta$ is a small constant introduced to prevent numeric overflow. A full derivation of $\xi$ and a sensitivity analysis of $\varepsilon$ can be found in Appendix D and Appendix F.9.

It is worth noting that, although we use the late fusion module $\mathcal{F}(\cdot)$ as the prediction head, we consider it a plug-and-play module, allowing $\hat{y}^{\mathcal{C}}$ to be calculated in any manner. This enables the application of our method to evaluate the contributions of modalities across various multimodal regression tasks like multimodal formation energy prediction.

**Contribution of a Single Modality $m$**   After the above analysis, we can evaluate the contributions of the modality subset $\mathcal{C}$. To assess the contribution of a single modality $m$, a straightforward approach is forward reasoning (Lynch & Vaandrager, 1995). This involves considering the modality subset $\mathcal{C}$ to contain only the modality $m$ under evaluation. Let $\widetilde{\beta}(m)$ represent the benefit of $m$, which is calculated with forward reasoning as follows:

$$\widetilde{\beta}(m) = \mathcal{B}(\mathcal{C}) = \mathcal{B}(\{m\}). \tag{12}$$

However, this approach encounters two significant issues:

1. The $\widetilde{\beta}(m)$, calculated by equation 12, is always greater than $0$. This implies that it cannot reflect the negative contribution of the modality to the output.

2. It calculates the contribution of each modality independently, ignoring the complementary information that exists between modalities, leading to an underestimation of the real contribution of modality $m$.

To accurately reflect the contribution of each modality, we introduce causal diagram (Knoblock, 1994) and counterfactual reasoning (Roese, 1997) to model the contribution of a single modality $m$. In the causal diagram, nodes represent causes and effects, while edges denote the influence from cause to effect. This diagram aids in understanding both the combinatorial and constraint relationships among each input condition, as well as the dependent relationships between causes and effects. Counterfactual reasoning, a common method in causal inference, explores cause-effect relationships by considering the outcomes of altering a cause, instead of directly examining the cause leading to an effect.

Figure 3 shows the causal diagram for a general multimodal task. To model the contribution of a single modality using counterfactual reasoning, we calculate the contributions of all modalities $\mathcal{B}(\mathcal{M})$. Subsequently, we exclude the modality $m$, which is under evaluation, from set $\mathcal{M}$, obtaining the remaining contribution $\mathcal{B}(\mathcal{M} \setminus \{m\})$. The contribution of modality $m$ is then defined as follows:

$$\beta(m) = \mathcal{B}(\mathcal{M}) - \mathcal{B}(\mathcal{M} \setminus \{m\}). \tag{13}$$

Leveraging the approach proposed in this subsection, we can quantitatively observe the modality preference issue existing in multimodal yield prediction. This aids in enhancing the explainability of observed phenomena by quantitatively representing the contributions of various modalities. A detailed pseudo-code implementation of the algorithm can be found in Appendix S.

### 3.3 MODEL TRAINING AND INFERENCE

Given $N$ reaction samples, each sample can be represented as $\boldsymbol{x}_i = \{\boldsymbol{x}_i^{\mathrm{s}}, \boldsymbol{x}_i^{\mathrm{g}}, \boldsymbol{x}_i^{\mathrm{f}}\}$. Our model aims to predict the mean $\mu(\boldsymbol{x}_i)$ and variance $\sigma(\boldsymbol{x}_i)$ of the yield distribution for sample $\boldsymbol{x}_i$. The actual yield of reaction sample $\boldsymbol{x}_i$ is denoted by $y_i$; whereas the predicted yield is expressed as $\hat{y}_i$; and we set $\hat{y}_i = \mu(\boldsymbol{x}_i)$ during training. To bridge the gap between $y_i$ and $\hat{y}_i$, we employ Mean Square Error (MSE) loss:

$$\mathcal{L}_{\mathrm{mse}} = \frac{1}{N} \sum_{i=1}^{N} \parallel \hat{y}_i - y_i \parallel^2 . \tag{14}$$

To mitigate the uncertainty in the predicted yield distribution arising from variables like molecular configuration and conformation, we introduce the uncertainty loss proposed in (Kendall & Gal, 2017), expressed as follows:

$$\mathcal{L}_{\mathrm{uct}} = \frac{1}{N} \sum_{i=1}^{N} \left( \frac{1}{2\sigma(\boldsymbol{x}_i)^2} \parallel \hat{y}_i - y_i \parallel^2 + \frac{1}{2} \log \sigma(\boldsymbol{x}_i)^2 \right) . \tag{15}$$

To promote cooperation among modalities, we employ the InfoNCE loss (He et al., 2020) during training to align the features captured from 2D graph ($\boldsymbol{h}_i^{\mathrm{g}}$) and fingerprints ($\boldsymbol{h}_i^{\mathrm{f}}$) with temperature parameter $\tau$:

$$\mathcal{L}_{\mathrm{info}} = -\frac{1}{2N} \sum_{i=1}^{N} \log \frac{\exp(\boldsymbol{h}_i^{\mathrm{g}} \cdot \boldsymbol{h}_{i+}^{\mathrm{f}}/\tau)}{\sum_{j=1}^{N} \exp(\boldsymbol{h}_i^{\mathrm{g}} \cdot \boldsymbol{h}_{j-}^{\mathrm{f}}/\tau)} - \frac{1}{2N} \sum_{i=1}^{N} \log \frac{\exp(\boldsymbol{h}_i^{\mathrm{f}} \cdot \boldsymbol{h}_{i+}^{\mathrm{g}}/\tau)}{\sum_{j=1}^{N} \exp(\boldsymbol{h}_i^{\mathrm{f}} \cdot \boldsymbol{h}_{j-}^{\mathrm{g}}/\tau)} .$$

To mitigate the issue of modal preference and enhance the predictive capacity of each modality individually, we introduce the following loss function:

$$\mathcal{L}_{\mathrm{prefer}} = \frac{1}{|\mathcal{M}| \cdot N} \sum_{m \in \mathcal{M}} \sum_{i=1}^{N} \parallel \hat{y}_i^m - y_i \parallel^2 , \tag{16}$$

where $\mathcal{M}$ denotes the set of all modalities employed and $\hat{y}_i^m$ signifies the yield predicted by single modality $m$. The goal of $\mathcal{L}_{\mathrm{prefer}}$ is to improve the ability of each modality $m$ to predict results independently. We provide more analysis on $\mathcal{L}_{\mathrm{prefer}}$ in Appendix N.

The overall loss is a sum of its components, balanced by hyperparameters $\lambda_{\mathrm{uct}}$, $\lambda_{\mathrm{info}}$ and $\lambda_{\mathrm{prefer}}$:

$$\mathcal{L} = \mathcal{L}_{\mathrm{mse}} + \lambda_{\mathrm{uct}} \mathcal{L}_{\mathrm{uct}} + \lambda_{\mathrm{info}} \mathcal{L}_{\mathrm{info}} + \lambda_{\mathrm{prefer}} \mathcal{L}_{\mathrm{prefer}} . \tag{17}$$

For inference, for a sample $\boldsymbol{x}_i$, by utilizing the reparameterization trick, we predict the yield as $\hat{y}_i = \mu(\boldsymbol{x}_i) + \epsilon * \sigma(\boldsymbol{x}_i)$, where $\epsilon$ is sampled from a standard normal distribution.

## 4 EXPERIMENT

### 4.1 EXPERIMENTAL SETUP

**Datasets** We utilized two classic yield prediction datasets: the Buchwald-Hartwig dataset (Ahneman et al., 2018) ($3,955$ reactions) and Suzuki-Miyaura dataset (Perera et al., 2018) ($5,760$ reactions). Both datasets include high-throughout experiments focused on cross-coupling reactions. Additionally, we incorporated the Amide Coupling Reaction (ACR) dataset (Lab, 2024) , which contains $41,239$ amide coupling reactions derived from Reaxys (Saebi et al., 2023). The ACR dataset is more challenging than the other two high-throughput experimental (HTE) datasets, characterized by its larger scale and greater diversity of reaction types.

**Baselines** We compared our method with both unimodal and multimodal yield prediction models. **One-hot** (Chuang & Keiser, 2018), **Yield-BERT** (Schwaller et al., 2020), **MPNN** (Kwon et al., 2022) and **DRFP** (Probst et al., 2022) predict the yield of a reaction using features extracted from a single modality. **YieldGNN** (Saebi et al., 2023), **ReaMVP** (Shi et al., 2024) and **UAM** (Chen et al., 2024) are recent yield prediction models that extract and fuse information from multiple modalities.

### 4.2 QUANTITATIVE EVALUATION OF PERFORMANCE

We evaluate our model against two categories of baselines: unimodal (One-hot, Yield-BERT, MPNN, DRFP) and multimodal (YieldGNN, ReaMVP, UAM). Table 1 summarizes the performance

Table 1: Results on Buchwald-Hartwig, Suzuki-Miyaura and Amide coupling reaction datasets.

| | MAE (↓) | | | RMSE (↓) | | | $R^2$ (↑) | | |
|---|---|---|---|---|---|---|---|---|---|
| | BH | SM | ACR | BH | SM | ACR | BH | SM | ACR |
| One-hot | $6.08_{(0.08)}$ | $8.55_{(0.08)}$ | - | $9.02_{(0.16)}$ | $12.27_{(0.15)}$ | - | $0.890_{(0.005)}$ | $0.809_{(0.023)}$ | - |
| Yield-BERT | $3.09_{(0.12)}$ | $6.60_{(0.27)}$ | $16.52_{(0.20)}$ | $4.80_{(0.26)}$ | $10.52_{(0.48)}$ | $21.12_{(0.13)}$ | $0.969_{(0.004)}$ | $0.859_{(0.012)}$ | $0.172_{(0.016)}$ |
| MPNN | $2.92_{(0.06)}$ | $6.12_{(0.22)}$ | $16.31_{(0.22)}$ | $4.43_{(0.09)}$ | $9.47_{(0.46)}$ | $20.86_{(0.27)}$ | $0.974_{(0.001)}$ | $0.886_{(0.010)}$ | $0.188_{(0.021)}$ |
| DRFP (MLP) | $3.69_{(0.05)}$ | $7.20_{(0.08)}$ | $16.15_{(0.17)}$ | $5.51_{(0.20)}$ | $10.78_{(0.18)}$ | $20.38_{(0.14)}$ | $0.960_{(0.003)}$ | $0.851_{(0.005)}$ | $0.207_{(0.010)}$ |
| YieldGNN | $3.89_{(0.14)}$ | $6.96_{(0.25)}$ | $15.27_{(0.18)}$ | $6.01_{(0.21)}$ | $11.00_{(0.37)}$ | $19.82_{(0.08)}$ | $0.953_{(0.003)}$ | $0.845_{(0.011)}$ | $0.216_{(0.013)}$ |
| ReaMVP | $3.31_{(0.8)}$ | $6.94_{(0.23)}$ | $16.02_{(0.16)}$ | $5.09_{(0.20)}$ | $10.53_{(0.33)}$ | $20.51_{(0.10)}$ | $0.966_{(0.004)}$ | $0.856_{(0.011)}$ | $0.201_{(0.019)}$ |
| UAM | $2.89_{(0.06)}$ | $6.04_{(0.18)}$ | $14.76_{(0.15)}$ | $4.36_{(0.10)}$ | $9.23_{(0.40)}$ | $19.33_{(0.10)}$ | $0.976_{(0.001)}$ | $0.888_{(0.009)}$ | $0.262_{(0.009)}$ |
| Harmony | $\mathbf{2.73}_{(0.09)}$ | $\mathbf{5.83}_{(0.10)}$ | $\mathbf{14.72}_{(0.17)}$ | $\mathbf{4.09}_{(0.13)}$ | $\mathbf{9.22}_{(0.31)}$ | $\mathbf{18.88}_{(0.10)}$ | $\mathbf{0.978}_{(0.001)}$ | $\mathbf{0.893}_{(0.007)}$ | $\mathbf{0.320}_{(0.008)}$ |

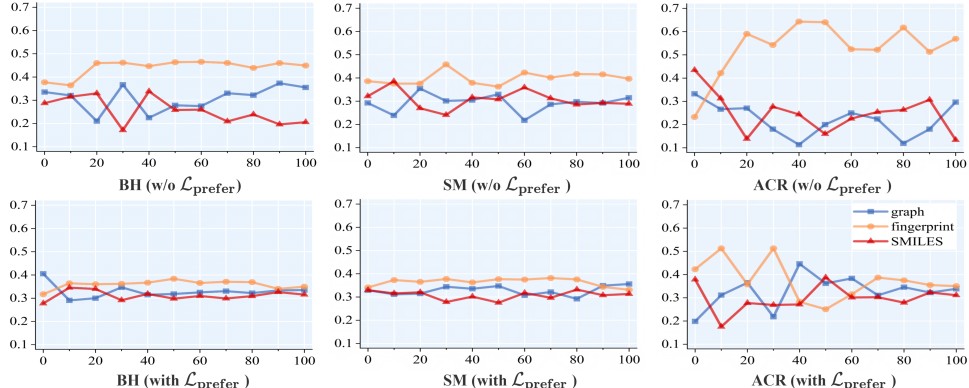

Figure 4: Evolution of normalized contributions from the three modalities over training epochs on the Buchwald-Hartwig, Suzuki-Miyaura and Amide coupling reaction datasets. The plots compare models trained with and without the $\mathcal{L}_{\mathrm{prefer}}$ loss.

across three datasets. Our model consistently outperforms all baselines across all metrics, demonstrating the effectiveness of our modality fusion strategy.

The analysis of our results highlights the advantages of multimodal fusion. While multimodal models generally outperform unimodal ones, as observed on the homogeneous HTE datasets, the true challenge lies in effectively integrating modalities for complex and diverse reactions. The performance on the ACR dataset exposes a key weakness in existing multimodal approaches: their fusion process is often imbalanced, preventing them from fully leveraging the available information. Our model overcomes this by enforcing balanced contributions from each modality, a design crucial for its superior performance. This leads to a new state-of-the-art, with statistical tests (Appendix L) confirming the significant improvements over strong baselines like UAM.

To comprehensively evaluate its generalization capability, we validated Harmony across a range of demanding benchmarks, where it not only achieved SOTA performance on large-scale datasets like USPTO but also surpassed all deep learning models on the noisy BH(ELN) dataset and demonstrated consistent superiority in highly challenging out-of-sample (OOS) tests, as detailed in Appendix O.

### 4.3 Qualitative Analysis of Modality Imbalance

A critical challenge in multimodal learning for reaction prediction is modality preference, where models over-rely on a single information source, especially for complex tasks. Our work is the first to identify and address this phenomenon in the context of yield prediction. As illustrated in Figure 4, this issue is particularly pronounced on the diverse ACR dataset, where the model exhibits a strong bias towards the fingerprint modality, neglecting other complementary sources. In contrast, contributions on simpler HTE datasets (BH and SM) are inherently balanced. Our proposed preference loss, $\mathcal{L}_{\mathrm{prefer}}$ directly rectifies this imbalance. It steers the highly skewed contributions towards a uniform distribution, validating Harmony's ability to enforce a balanced and effective fusion of all modalities, even for the most challenging reactions.

### 4.4 The Effectiveness of the Hierarchical Architecture And $\mathcal{L}_{\mathrm{PREFER}}$

Harmony attains state-of-the-art (SOTA) performance by virtue of its hierarchical fusion framework, which adeptly captures cross-modal interactions at various levels, and the modal contribution balancing objective $\mathcal{L}_{\mathrm{prefer}}$, which mitigates over-dependence on any single modality.

Table 4: Ablation Results on Buchwald-Hartwig, Suzuki-Miyaura and ACR datasets.

| | MAE (↓) | | | RMSE (↓) | | | $R^2$ (↑) | | |
|---|---|---|---|---|---|---|---|---|---|
| | BH | SM | ACR | BH | SM | ACR | BH | SM | ACR |
| w/o $\mathcal{L}_{\text{prefer}}$ | - | - | 14.77 | - | - | 19.45 | - | - | 0.278(−13.1%) |
| w/o $\mathcal{L}_{\text{info}}$ | 2.97 | 6.24 | 15.22 | 4.65 | 9.89 | 19.47 | 0.971(−0.7%) | 0.875(−20.2%) | 0.276(−15.9%) |
| w/o molecular-level fusion | 2.93 | 5.99 | 15.09 | 4.47 | 9.53 | 19.06 | 0.974(−0.4%) | 0.884(−10.1%) | 0.306(−4.4%) |
| w/o KAN | 3.13 | 6.34 | 14.76 | 5.08 | 10.03 | 18.91 | 0.966(−1.2%) | 0.871(−26.6%) | 0.317(−0.9%) |
| w/o Mamba | 3.14 | 6.13 | 14.98 | 4.99 | 9.59 | 19.06 | 0.967(−1.1%) | 0.882(−12.3%) | 0.306(−4.4%) |
| w/o Seq | 2.89 | 6.06 | 14.79 | 4.61, | 9.53 | 19.02 | 0.972(−0.6%) | 0.884(−10.1%) | 0.310(−3.1%) |
| w/o Graph | 2.91 | 6.20 | 15.38 | 4.55 | 9.81 | 19.47 | 0.973(−0.5%) | 0.877(−17.9%) | 0.276(−15.9%) |
| w/o ECFPs | 5.13 | 7.12 | 16.75 | 7.48 | 10.59 | 20.82 | 0.926(−5.3%) | 0.856(−41.4%) | 0.173(−45.9%) |
| Harmony | **2.73** | **5.83** | **14.72** | **4.09** | **9.22** | **18.88** | **0.978** | **0.893** | **0.320** |

We conduct ablation studies to validate Harmony's core components. First, retrofitting the UAM baseline with our hierarchical framework boosted performance by a substantial 10% at a negligible cost of 1.6% additional parameters (Table 2). Second, applying our parameter-free balancing loss to UAM independently yielded a rela-

Table 2: Experiments on improving UAM with $\mathcal{L}_{\text{prefer}}$ and hierarchical mechanism.

| Datasets | Amide Coupling Reaction(ACR) | | | |
|---|---|---|---|---|
| Metrics | MAE(↓) | RMSE(↓) | $R^2$(↑) | trainable parameters(↓) |
| UAM | 14.76 | 19.33 | 0.262 | 37,700,166 |
| A hierarchical UAM | **14.66** | 19.31 | 0.288 | 38,297,511 |
| UAM with $\mathcal{L}_{\text{prefer}}$ | 14.69 | 19.47 | 0.277 | 37,700,166 |
| Harmony | 14.72 | **18.88** | 0.320 | 12,591,530 |

tive improvement of approximately 6% in $R^2$, demonstrating its effectiveness as a model-agnostic component. Finally, the complete Harmony model maintains a smaller parameter footprint than the baseline by freezing its SMILES encoder, which enhances both training speed and stability.

## 4.5 RATIONALE FOR THE HIERARCHICAL FUSION STRATEGY

Our analysis, illustrated in Figure 3, reveals that the specific hierarchy of our fusion strategy is critical. For instance, fusing fingerprints with either 2D graphs or SMILES prematurely introduces high-level abstractions that can obscure essential, fine-grained structural details or lack complementary information. The optimal ap-

Table 3: Ablation study on the early-stage fusion strategy on the ACR dataset.

| Fused Modality | MAE(↓) | RMSE(↓) | $R^2$(↑) |
|---|---|---|---|
| SMILES+Fingerprints | 15.35(0.20) | 19.62(0.15) | 0.265(0.009) |
| 2D graph+Fingerprints | 14.97(0.20) | 19.08(0.11) | 0.293(0.010) |
| **Harmony(SMILES+2D graph)** | **14.72(0.17)** | **18.88(0.10)** | **0.320(0.008)** |

proach, therefore, is to first fuse SMILES and 2D graphs. This initial step integrates sequential and topological information to build a comprehensive molecular representation, which then provides a solid foundation for the final fusion with abstract reaction features from the fingerprints.

## 4.6 ABLATION STUDIES

We conducted ablation studies on the BH, SM, and ACR datasets (Table 4). Our proposed losses are indispensable: removing either $\mathcal{L}_{\text{prefer}}$ or $\mathcal{L}_{\text{info}}$ consistently degrades performance, confirming their necessity for mitigating modality bias and enabling effective fusion. The architectural advantages are equally clear: eliminating the hierarchical design and directly fusing all modalities causes a significant performance drop, while replacing the KAN head with an MLP or the Mamba encoder with a Transformer also results in notable declines. These findings validate the effectiveness of KAN for non-linear mapping and highlight Mamba's intrinsic benefits in processing fingerprint data.

In modality ablation, omitting any single modality causes a substantial performance decline, indicating that Harmony integrates the information of each modality comprehensively for yield prediction. Moreover, the magnitude of decline aligns with the learned modality contributions (Figure 4), fingerprints being the most impactful, which validates our modality evaluation approach.

## 5 CONCLUSION

In this work, we present Harmony, an efficient hierarchical multimodal fusion framework for reaction yield prediction. Harmony not only achieves state-of-the-art performance on benchmark datasets but also demonstrates strong generalization across diverse reaction types. Its effectiveness stems from the synergy of hierarchical fusion, which prevents destructive interference between fine-grained molecular features and high-level reaction representations. Equipped with a plug-and-play contribution evaluation module and a preference-balancing optimization objective, Harmony effectively mitigates modality collapse and enhances the robustness of yield predictions. Beyond chemistry, the core design principles of Harmony, hierarchical fusion and preference balancing, are broadly applicable to multimodal regression tasks in materials science and biology, making it valuable to both the chemistry and machine learning communities.

## REPRODUCIBILITY STATEMENT

To ensure result reproducibility, key resources and details are referenced as follows:

**Code**  The Harmony framework (hierarchical fusion, modality assessment, prefer-balancing objective) is available at: `https://anonymous.4open.science/r/F6BB`, covering all modules in Section 3. We provide complete code, training and evaluation scripts, partial datasets (the ACR dataset is not included, as access to it requires a Reaxys database subscription), as well as model weights and hyperparameter configuration files, ensuring the reproducibility of the experimental results.

**Datasets**  Benchmark datasets (Buchwald-Hartwig, Suzuki-Miyaura, Amide Coupling Reaction) are described in Section 4.1 and Appendix F.3; the Amide dataset is linked at `https://github.com/isayevlab/amide_reaction_data`. Data partitioning (Appendix F.2) and modality construction (Appendix F.5) are detailed.

**Experimental Settings**  Implementation details (optimizer, epochs, batch size, learning rate) in Appendix F.1; hyperparameters (e.g., $\lambda_{uct}$, $\lambda_{prefer}$) and tuning in Appendix F.8; metrics (MAE, RMSE, $R^2$) in Appendix E; statistical tests in Appendix L.

**Methodological Transparency**  Smoothing factor $\xi$ derivation (Appendix D), modality assessment pseudocode (Appendix S), fingerprint design (Appendix F.6); ablation studies (Section 4.6) and sensitivity analysis (Appendix F.9) validate robustness, with results averaged over 10 shuffles (Appendix F.1).

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

## A  THE USE OF LARGE LANGUAGE MODELS (LLMs)

During the research process, we mainly used large language models (LLMs) in two aspects. First, during the manuscript writing phase, we utilized LLMs to polish the language of the article, and we conducted strict verification on the content they generated. Second, in Appendix H, for the experiment that introduces the additional modality of large language model embedding into Harmony, we used the open-source Qwen3-Embedding-0.6B model.

## B  EXTENDED DISCUSSION OF CONTRIBUTIONS

This section provides a more detailed exposition of Harmony's primary contributions, elucidating the methodological innovations and their synergistic interplay. Our work introduces a principled and empirically validated framework that advances reaction yield prediction by addressing foundational challenges in multi-scale chemical modeling.

### B.1  A CHEMISTRY-AWARE HIERARCHICAL FUSION ARCHITECTURE

A core contribution of Harmony is its departure from the prevailing "flat fusion" paradigm. We introduce a tiered fusion architecture designed to mirror the natural hierarchy of chemical information. This principled design is guided by the physicochemical concept that molecular-level structural and topological information must first be consolidated into a coherent representation of the chemical entities involved (i.e., reactants and products) before being integrated with abstract, reaction-level mechanistic patterns.

Specifically, our architecture performs an early fusion of SMILES (sequential) and 2D graph (topological) representations to form robust, unified molecular embeddings. Only after this consolidation does the model proceed to a late fusion stage, integrating these molecular embeddings with high-level reaction fingerprints. This staged, granularity-aware information flow is not merely an engineering choice; it is a crucial mechanism to prevent the information dilution and feature interference endemic to flat architectures. As demonstrated in our ablation studies (Table 4), this specific, chemistry-aware implementation provides a significant performance advantage over simply adapting a baseline model to a hierarchical structure, highlighting the non-triviality and importance of our design.

### B.2  A PRINCIPLED FRAMEWORK FOR DIAGNOSING AND BALANCING MODALITY CONTRIBUTIONS

To address the critical issue of representational overshadowing, we developed a novel methodological framework that moves from quantitative diagnosis to targeted optimization. To our knowledge, this represents the first application of such a systematic, data-driven approach to fusion control in the context of multimodal AI for chemistry.

1. **Quantitative Diagnosis via Counterfactual Assessment:** We introduce a quantitative framework based on counterfactual reasoning to measure the net contribution of each modality within the end-to-end model. This moves beyond the limitations of conventional, post-hoc ablation studies by providing a precise, in-situ assessment of how each representation dynamically influences the final prediction.

2. **Targeted Optimization via a Preference-Balancing Objective:** The insights gained from our contribution assessment directly inform the design of a differentiable, plug-and-play preference-balancing objective. This objective actively counteracts modality dominance by encouraging each representation to maintain its independent predictive utility. The synergy between our diagnostic tool and this targeted optimization creates a principled, verifiable mechanism for enforcing a balanced and robust fusion process.

### B.3  REACTION FINGERPRINTS AS DYNAMIC CHANGE ENCODERS

We advance reaction-level feature engineering with a novel fingerprint design that more directly captures the dynamics of a chemical transformation. Unlike classic methods such as the Differential

Reaction Fingerprint (DRFP), which rely on a symmetric difference (a binary presence/absence metric), our approach computes bit-wise differences of summed reactant and product fingerprints. This method preserves not only the location of structural changes but also their magnitude and direction (e.g., the formation of multiple instances of a substructure).

This representation provides a richer, more mechanistically relevant signal that reflects the underlying events of bond-breaking and forming. Our gradient analyses confirm that the model learns to attend specifically to these high-signal change-bits (Figure 8), and comparative experiments demonstrate the superior effectiveness of this approach over DRFP (Table 8).

### B.4 Synthesis: An Integrated Framework for Robust Generalization

The ultimate strength of Harmony lies in the synergistic integration of these three innovations. The hierarchy-aware architecture provides the necessary structural foundation for multi-scale modeling. The causal assessment and balancing framework then ensures that information flows through this structure in a robust and equitable manner. Finally, the dynamic change-aware fingerprints supply the model with a high-fidelity signal of the reaction's core transformation.

By combining these elements, Harmony systematically addresses the challenge of representational overshadowing. This integrated design is directly responsible for its robust generalization capabilities, particularly on molecules absent from the training set, and its state-of-the-art performance across three widely-used benchmarks. This work thus represents a significant step towards developing more reliable and practical AI tools for chemical synthesis planning.

## C Notation Tables

To improve the readability of this paper, we provide notations tables for the notations defined in this paper.

### C.1 Notation Table for Section 3.1

Table 5 shows all the symbols we have used in Section 3.1. In this section, the relationship among the unordered set of all modalities $\mathcal{M}$, an individual modality $m$, and molecular modalities $s, g, f$ is as follows: $\mathcal{M} = \{m | m \in \{s, g, f\}\}$.

### C.2 Notation Table for Section 3.2

Table 6 shows the symbols that we have newly defined in Section 3.2. Some of these symbols are also presented in Table 5 and will not be repeated here. To simplify the representation in Section 3.2, we removed the subscript $i$; for example, $x_i$ was simplified to $x$.

## D Derivation of the Smoothing Factor $\xi$

We define $\hat{y}^{\mathcal{C}}$ as the yield predicted using the subset of modalities $\mathcal{C}$, and $y$ as the actual yield. According to Equation equation 10, the subset of modalities $\mathcal{C}$ contributes positively to the outcome if and only if $|\hat{y}^{\mathcal{C}} - y| \leq \varepsilon$, where $\varepsilon$ is a threshold defined by consideration, generally satisfies $\varepsilon > 0$. Therefore, we obtain:

$$0 \leq \| \hat{y}^{\mathcal{C}} - y \|^2 \leq \varepsilon^2, \ \varepsilon > 0.$$

Subsequently, we derive the following:

$$0 \leq \frac{\| \hat{y}^{\mathcal{C}} - y \|^2}{\varepsilon^2} \leq 1, \ \varepsilon > 0. \tag{18}$$

We need to find a continuous function such that the modal contribution increases as the term $\frac{\|\hat{y}^{\mathcal{C}} - y\|^2}{\varepsilon^2}$ in Equation equation 18 approaches 0, and decreases as this term approaches 1. We discovered that the $-\log(\cdot)$ function satisfies this property within the domain of 0 to 1, hence we define:

Table 5: The table of notations used in Section 3.1.

| Notation | Description |
|---|---|
| $N$ | The number of all reaction samples. |
| $\boldsymbol{x}_i$ | The $i$-th sample among all reaction samples where $i \in \{1, \ldots, N\}$. |
| $y_i$ | The ground-truth reaction yield of $i$-th sample. |
| $\mathcal{M}$ | The unordered set of all modalities. |
| $m$ | A modality in $\mathcal{M}$. |
| $s, g, f$ | Molecular modalities, representing SMILES, 2D graphs and fingerprints, respectively. |
| $T_i$ | The types of molecules involved in the $i$-th reaction sample. |
| $\boldsymbol{x}_i^s, \boldsymbol{x}_i^f$ | Data of SMILES and fingerprints modalities for the $i$-th sample. |
| $\boldsymbol{x}_{i,j}^g$ | Data of the 2D graph modality for the $i$-th sample's $j$-th molecule, where $j \in \{1, \ldots, T_i\}$. |
| $\boldsymbol{h}_{i,j}^g$ | Encoded features of the 2D graph modality for the $i$-th sample's $j$-th molecule. |
| $\widetilde{\boldsymbol{h}}_{i,j}^g$ | Encoded features of the 2D graph modality integrated with global reaction information. |
| $\boldsymbol{h}_i^s, \boldsymbol{h}_i^g, \boldsymbol{h}_i^f$ | Features extracted from each molecular modality. |
| $\boldsymbol{h}_i^m$ | Features extracted from modality $m$. |
| $\hat{y}_i$ | The model prediction yield obtained through reparameterization trick. |
| $\epsilon$ | A random value sampled from a standard normal distribution. |
| $\mu(\boldsymbol{x}_i)$ | The mean of the yield distribution for the $i$-th reaction. |
| $\sigma(\boldsymbol{x}_i)$ | Standard deviation of the yield distribution for the $i$-th reaction. |
| $\text{Enc}^s(\cdot), \text{Enc}^g(\cdot), \text{Enc}^f(\cdot)$ | Encoders for each molecular modality data. |
| $\text{LBRD}(\cdot)$ | Corresponding LBRD module. |
| $\mathcal{E}(\cdot)$ | Embedding layer for processing fingerprints data. |
| $\text{KAN}(\cdot)$ | Corresponding KAN layer. |
| $\mathcal{F}(\cdot)$ | Corresponding late fusion module. |

$$\xi = -\log \frac{\| \hat{y}^{\mathcal{C}} - y \|^2}{\varepsilon^2}$$
$$= 2 \cdot \log \frac{\varepsilon}{|\hat{y}^{\mathcal{C}} - y|}, \ \varepsilon > 0. \tag{19}$$

The definition above encounters two problems: the first is that the range of $\xi$ is from 0 to $+\infty$. To avoid excessive contributions from certain samples, which could disrupt the entire evaluation process, we aim to limit the value range between 0 and 1. The second problem is that the numerator $|\hat{y}^{\mathcal{C}} - y|$ may equal 0, to prevent numerical underflow, we define a very small number $\delta$, which in our implementation is valued at $10^{-5}$. Therefore, our final definition of the smoothing factor $\xi$ is:

$$\xi = \min \left( 1, 2 \cdot \log \frac{\varepsilon}{|\hat{y}^{\mathcal{C}} - y| + \delta} \right), \ \varepsilon > 0. \tag{20}$$

Equation 20 can be represented as a piecewise function:

$$\xi = \begin{cases} 1 & 0 \le |\hat{y}^{\mathcal{C}} - y| \le \dfrac{\varepsilon}{\sqrt{\mathrm{e}}}, \\ 2 \cdot \log \dfrac{\varepsilon}{|\hat{y}^{\mathcal{C}} - y| + \delta} & \dfrac{\varepsilon}{\sqrt{\mathrm{e}}} < |\hat{y}^{\mathcal{C}} - y| \le \varepsilon . \end{cases}$$

Table 6: The table of notations used in Section 3.2.

| Notation | Description |
|---|---|
| $n$ | Number of modalities in $\mathcal{M}$ |
| $\mathcal{C}$ | Modality subset which contains modalities that will be evaluated. We have $\mathcal{C} \subset \mathcal{M}$. |
| $\boldsymbol{x}$ | A sample among all reaction samples. |
| $\boldsymbol{x}^m$ | Data of modality $m$ of a sample $x$. |
| $y$ | The groud-truth yield of reaction sample $x$. |
| $\hat{y}$ | The predicted yield of the model using all modalities in $\mathcal{M}$. |
| $\hat{y}^{\mathcal{C}}$ | The predicted yield of the model using a modality subset $\mathcal{C}$. |
| $\varepsilon$ | Threshold for positive contributions from the evaluated modalities. |
| $\xi$ | Smoothing factor for modality contributions. |
| $\delta$ | A small constant to prevent numerical overflow. |
| $\mathcal{T}(\cdot\,;\cdot\,)$ | Mapping function that returns different results based on whether the modality $m$ is included in $\mathcal{C}$. |
| $\Phi^m(\cdot)$ | Feature extractor for the data of modality $m$. |
| $\mathcal{B}(\cdot)$ | The contribution of the modality set to the model's prediction results. |
| $\beta(\cdot)$ | The contribution of a single modality to the model's prediction results. |

This indicates that our smoothing factor $\xi$ also has a threshold $\frac{\varepsilon}{\sqrt{\mathrm{e}}}$. It acts like a "passing line"; contributions exceeding this line are directly considered to have a coefficient of 1. The existence of the passing line is similar to the idea of reparameterization, where we reserve a certain margin of error for the prediction results.

## E  EVALUATION METRICS

We evaluate Harmony using the most commonly used evaluation metrics in regression tasks, including mean absolute error (MAE), root mean squared error (RMSE), and R-Square ($R^2$). First, MAE directly calculates the mean of the absolute differences between the predicted results and the true labels for all samples. Its numerical range is from 0 to $+\infty$, and the closer the value is to 0, the smaller the gap between the model's predictions and the actual values, indicating better model performance. We adopt the same notation as in the main context to formally define our evaluation metrics. If we have $N$ samples, with the true label of each sample being $y_i$ and the predicted result denoted as $\hat{y}_i$, then MAE can be defined as:

$$\mathrm{MAE} = \frac{1}{N} \sum_{i=1}^{N} |\hat{y}_i - y_i|.$$

RMSE calculates the square root of the average of the squared differences between the predicted results and the true labels for all samples. Its numerical range is also from 0 to $+\infty$, and the closer the value is to 0, the smaller the gap between the model's predictions and the actual values, indicating better model performance. RMSE can be defined as:

$$\mathrm{RMSE} = \sqrt{\frac{1}{N} \sum_{i=1}^{N} \parallel \hat{y}_i - y_i \parallel^2}$$

$R^2$ is a coefficient of determination, with its value ranging from a maximum of $1$ to a minimum of $0$. The closer the value is to $1$, the better the independent variables explain the dependent variable in regression analysis, thereby indicating a better model. The closer the value is to $0$, the worse the model. We define the average of the true values of the samples as $\overline{y}$, thus $R^2$ can be calculated as follows:

$$R^2 = 1 - \frac{\sum_{i=1}^{N} \| \hat{y}_i - y_i \|^2}{\sum_{i=1}^{N} \| \overline{y}_i - y_i \|^2}$$

## F  EXPERIMENTAL DETAILS

### F.1  IMPLEMENTATION DETAILS

The network is optimized using a AdamW optimizer (Loshchilov, 2017) and is trained for 300 epochs with a batch size of $128$. We adopt an initial learning rate of $5 \times 10^{-3}$ and use a cosine learning rate delay scheduler (Loshchilov & Hutter, 2016). The threshold for modality contribution is set to $\varepsilon = 0.1$. We adjust hyper-parameters of the loss function to $\lambda_{\text{uct}} = \lambda_{\text{info}} = 0.1, \lambda_{\text{prefer}} = 0.2$ for the ACR dataset and to $\lambda_{\text{uct}} = \lambda_{\text{info}} = 0.1, \lambda_{\text{prefer}} = 0$ for the other two datasets. We conduct a detailed analysis of the selection values of these hyper-parameters and their robustness in Appendix F.8. The model was built using the PyTorch framework and trained on an NVIDIA RTX A6000 GPU. It utilized mean absolute error (MAE), root mean squared error (RMSE), and R-Square ($R^2$) as metrics. To ensure fairness, the results were averaged over 10 random shuffles, along with their standard deviations.

### F.2  DATASET DETAILS

Table 7 presents basic information on the three benchmark reaction yield prediction datasets we have used, including the name, type, and number of samples in each dataset. Among these, the "High-throughput experiments dataset" refers to datasets where yield data are obtained through high-throughput experiments. This type of dataset contains fewer chemical reactions, and the reactions are similar in type and conditions, providing a good premise for neural network models to predict yields.

On the other hand, the "Large literature dataset" refers to samples from large databases, such as Reaxys. This type of dataset contains a large variety of chemical reactions with diverse reaction types and conditions, making it challenging for yield prediction models to achieve good results. Our next step is to try incorporating reaction conditions as additional components into the yield prediction model. This approach will help the model to integrate more information for more accurate yield predictions

Table 7: Basic descriptions of three reaction datasets.

| Dataset | Type | Reaction Number |
|---|---|---|
| Buchwald-Hartwig reaction | High-throughput experiments dataset. | $3,955$ |
| Suzuki-Miyaura reaction | High-throughput experiments dataset. | $5,760$ |
| Amide coupling reaction | Large literature dataset. | $41,239$ |

Furthermore, we also analyzed the distribution of the yields in various datasets, as shown in Figure 5. Each dataset has its unique yield distribution characteristics, but overall, the yield data are relatively uniform. For the BH dataset, over $40\%$ of the yield values fall between $0\%$ and $20\%$. For the SM dataset, approximately $35\%$ of the yield values are in the range of $10\%$ to $30\%$. For the ACR dataset, more than half of the samples have yield values between $60\%$ and $90\%$.

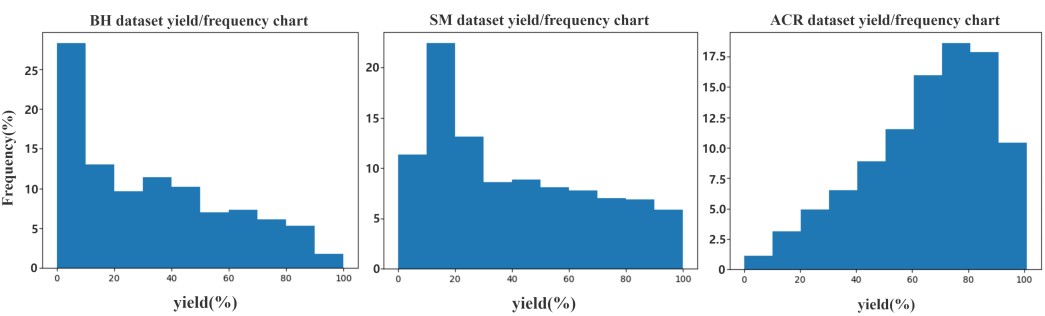

Figure 5: Yield distribution of three reaction datasets.

## F.3 DATASET PARTITIONING

To maintain a fair comparison with the baseline methods, we followed their partitioning approach to divide the data. For the BH and SM datasets, we split the datasets into training and testing sets with a ratio of $7 : 3$. For the ACR dataset, we divided it into training, validation, and testing sets with a ratio of $6 : 2 : 2$. Our ten-fold cross-validation experiments independently partitioned the data with each run. The ablation studies were conducted on pre-partitioned datasets for training and testing.

## F.4 RESOURCE CONSUMPTION FOR MODEL TRAINING

We used a single NVIDIA RTX A6000 48G GPU for model training. Our model was trained for 300 epochs on the train set of each dataset. It took approximately 4 hours to train on the BH dataset, about 8 hours on the SM dataset, and around 26 hours on the ACR dataset.

## F.5 CONSTRUCTION EACH MODALITY DATA

For data in the SMILES modality, both reactants and products involved in the reaction are represented as sequences of atomic strings. For example, the SMILES representation of the 2-methyl-N-(4-methylcyclohexyl)-1H-indole-3-carboxamide molecule can be expressed as "CC1CCC(CC1)NC(=O)C1=C(C)NC2=C1C=CC=C2". For chemical reactions in SMILES, we concatenate the SMILES of the reactants and products using the "." symbol. Finally, they are connected by the ">>" symbol.

For data in the 2D graph modality, we represent each reactant and product involved in the reaction as a molecular graph. For example, in the reaction to synthesize the 2-methyl-N-(4-methylcyclohexyl)-1H-indole-3-carboxamide molecule, the three molecules involved in the reaction are represented as shown in Figure 6. For each molecule, the features of the nodes in the molecule form a two-dimensional matrix, where the size of the first dimension is the number of atoms other than hydrogen atoms in the molecule, and the size of the second dimension is the feature dimension of each atom. The features of the bonds (edges) in the molecule are also represented as a two-dimensional matrix, where the size of the first dimension is the number of bonds other than those involving hydrogen atoms, and the second dimension is the feature dimension of the bonds. In addition, there is a sparse adjacency matrix used to represent which atoms are connected by these bonds.

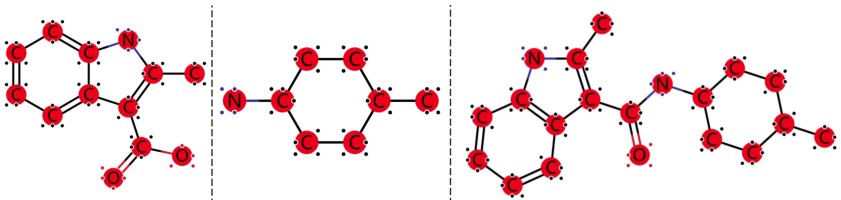

Figure 6: 2D graphs of the chemical reaction to synthesize the 2-methyl-N-(4-methylcyclohexyl)-1H-indole-3-carboxamide molecule.

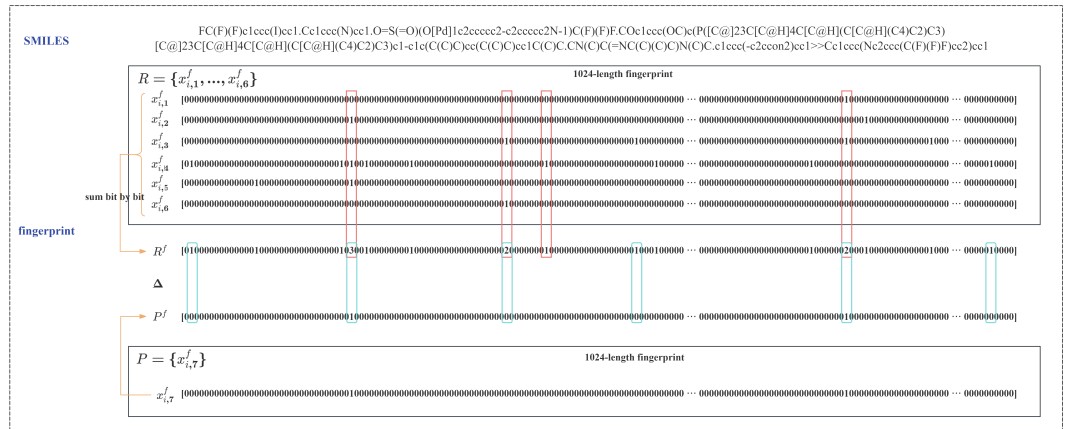

Figure 7: The calculation process of the chemical reaction and its fingerprint differences. The fingerprints of the reactants and products are represented by $R^f$ and $P^f$ respectively. Some positions for calculating the sum of the fingerprints of the reactants are marked with a red frame. Some positions of the fingerprint differences between the reactants and products are marked with a blue frame.

For data in the fingerprints modality, in Harmony, it is a sequence of $1024$ bits where each bit indicates whether the molecule has a certain structure or property. Since the number of reactants and products in each reaction is different, we cannot simply concatenate them. Therefore, we add up the fingerprints of the reactants and products separately, and then concatenate them along the feature dimension. This results in a sequence of 1024 pairs, where each pair is a numerical token representing the properties of the reactants and products, respectively. This design facilitates the use of embedding and natural language processing techniques for sequence feature extraction in later stages.

### F.6 EXTRACTING REACTION-LEVEL FEATURES IN MOLECULAR FINGERPRINTS

We would like to begin by introducing the differential reaction fingerprint (DRFP)(Probst et al., 2022), a well-established method that encodes the global information of chemical reactions. Following this, we will present our fingerprint design methodology. Finally, comparative experiments demonstrate that our fingerprint design outperforms DRFP in capturing the global information of chemical reactions.

In the DRFP algorithm, circular substructures are extracted from the molecules formed by combining reactants and reagents in the reaction SMILES, resulting in a set of molecular n-grams R. At the same time, a set of molecular n-grams P is extracted from the product molecules. Through calculating the symmetric difference of these two sets $S = R\Delta P$, the structural difference information between the reactants and the products can be obtained. This difference information is the key data used by DRFP for reaction classification and yield prediction. Subsequently, the symmetric difference set will be hashed and folded to be transformed into a fixed-length binary vector.

In Harmony, we also generate corresponding fingerprint data based on the SMILES of reactants and products. Suppose the reactant fingerprint set of chemical reaction is $R = \{\boldsymbol{x}^f_{i,1}, \ldots \boldsymbol{x}^f_{i,m}\}$, and the product fingerprint set is $P = \{\boldsymbol{x}^f_{i,m+1}\}$, where each molecular fingerprint data has a fixed length of 1024. We sum the fingerprints in the reactant fingerprint set R bit by bit to obtain a 1024-length fingerprint $R^f$, which embodies the structural information of the reactants. The product fingerprint set is processed in the same way to get another 1024-length fingerprint $P^f$. Instead of calculating the symmetric difference like DRFP, we propose to directly let the model learn the change amount at the corresponding position. For example, if the $i$-th bit of $R^f$ is $a$ and the $i$-th bit of $P^f$ is $b$, then the fingerprint change amount $\Delta = b - a$. In Figure 7, the positions of some fingerprint changes are marked with blue frames.

Similar to DRFP, our design enables the model to capture the structural difference information between reactants and products based on this fingerprint difference, and thus obtain the global infor-

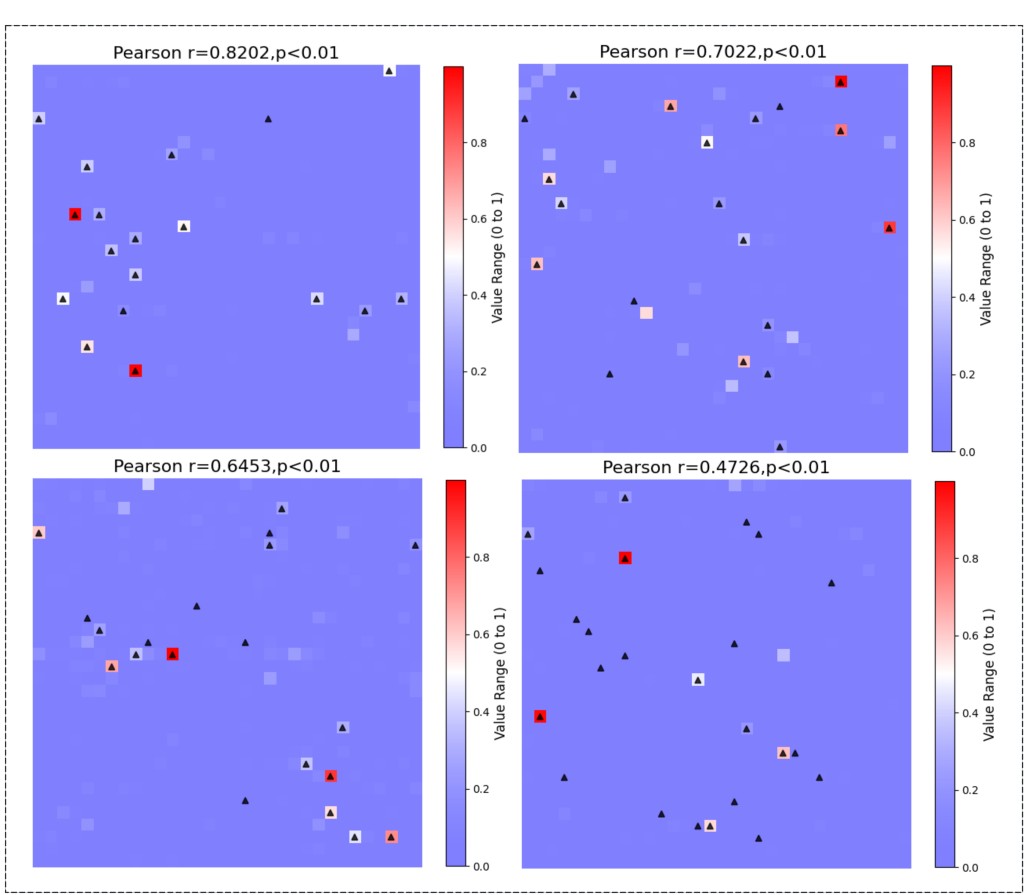

Figure 8: Fingerprint changes and gradient significance maps. The fingerprint of length 1024 is split into a 32x32 matrix. Black triangles indicate the positions of changes between the fingerprints of reactants and products. The closer the background color is to red, the more significant the gradient, meaning the model pays more attention to these positions. The Pearson correlation coefficient indicates a significant correlation between the two.

Table 8: The comparison between DRFP and our fingerprint difference method.

| | | BH | | | SM | | | ACR | | |
|---|---|---|---|---|---|---|---|---|---|---|
| | trainable parameters($\downarrow$) | MAE ($\downarrow$) | RMSE ($\downarrow$) | $R^2$ ($\uparrow$) | MAE ($\downarrow$) | RMSE ($\downarrow$) | $R^2$ ($\uparrow$) | MAE ($\downarrow$) | RMSE ($\downarrow$) | $R^2$ ($\uparrow$) |
| DRFP(mlp) | $561,601$ | 3.67 | 5.50 | 0.960 | 7.21 | 10.88 | 0.848 | 16.11 | 20.32 | 0.211 |
| Ours | $\mathbf{559,745}$ | **3.50** | **5.33** | **0.962** | **7.10** | **10.68** | **0.854** | **14.76** | **19.53** | **0.272** |

mation of the chemical reaction. Figure 8 show that the model's decision-making during the training process significantly depends on this fingerprint change amount.

In contrast to DRFP, our design does not include hashing and folding operations. This allows our fingerprints to retain more comprehensive information about chemical reactions. For example, it avoids the information loss caused by hash collisions. We compared the performance of our fingerprints with that of DRFP on three benchmark datasets(Table 8). Considering differences in data dimensions, the number of model parameters we used is not exactly the same. As demonstrated by the experimental results, our fingerprint design can more effectively represent the global information of chemical reactions compared with DRFP and achieves better results.

**Why do fingerprints impact model predictions more than other representations?** The pronounced influence of the reaction fingerprint modality is attributable to a fundamental dichotomy in both its encoded information and its strategic placement within our hierarchical architecture. First, a distinction exists in the nature of the encoded information. SMILES and 2D graphs function as static, molecular-level descriptors, representing the discrete states of reactants and products. From these, the model must implicitly infer the transformation process. Conversely, the differential fingerprint operates at a higher level of abstraction. It explicitly encodes the net structural transformation, providing a holistic, dynamic descriptor of the reaction itself—a direct and powerful signal for the predictive task.

Second, our architecture is deliberately designed to mirror this informational hierarchy. The model first consolidates the complementary molecular-level representations (SMILES and 2D graph) to construct a robust substrate descriptor. Only then is this integrated with the overarching, reaction-level context provided by the fingerprint. This design choice is empirically validated, as alternative strategies involving the premature fusion of the high-level fingerprint with either low-level modality were found to be suboptimal. Such early fusions lead to either an irrecoverable deficit of topological information (when fusing with SMILES) or inefficient feature blending and semantic overshadowing (when fusing with the 2D graph), ultimately confirming the efficacy of our hierarchical approach (Table 3).

## F.7 HYPERPARAMETER SETTINGS

We give the description of the hyperparameters in Table 9 and their values for each benchmark dataset in Table 10.

Table 9: Descriptions of the hyperparameters.

| **Hyperparameter** | **Description** |
|---|---|
| $\lambda_{\text{prefer}}$ | The weight of $\mathcal{L}_{\text{prefer}}$. |
| $g_{\text{hidden\_size}}$ | Size of hidden features for the 2D graph encoder (*i.e.*, GNN). |
| $g_{\text{num\_step\_mp}}$ | The number of message passing iterations between GNN nodes. |
| $g_{\text{num\_step\_set2set}}$ | The number of rounds for GNN to aggregate node information. |
| $f_{\text{dropout\_ratio}}$ | Dropout ratio for fingerprints encoder. |

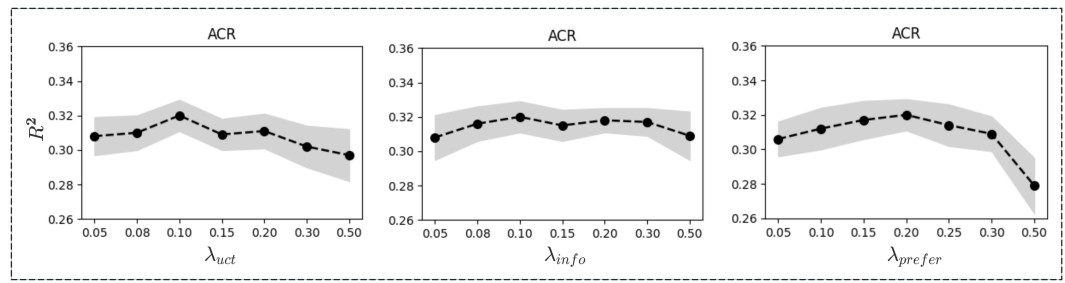

Figure 9: Parameter tuning process of critical parameters $\lambda_{uct}$, $\lambda_{info}$, and $\lambda_{prefer}$ conducted on the ACR dataset.

Table 10: Hyperparameters for Harmony in three benchmark datasets.

| Hyperparameter | BH | SM | ACR |
|---|---|---|---|
| $\lambda_{\text{prefer}}$ | 0 | 0 | 0.2 |
| $g_{\text{hidden\_size}}$ | 64 | 64 | 32 |
| $g_{\text{num\_step\_mp}}$ | 2 | 2 | 1 |
| $g_{\text{num\_step\_set2set}}$ | 3 | 3 | 1 |
| $f_{\text{dropout\_ratio}}$ | 0.25 | 0.1 | 0.25 |

F.8 HYPERPARAMETER TUNING PROCESS

We provided candidate hyperparameters (e.g. $\{0.05, 0.10, 0.15, 0.20, 0.25, 0.30, 0.50\}$ for $\lambda_{prefer}$) and determined the optimal via experiments, with all others similarly selected using the same method. We present the tuning process for three key hyperparameters to select them and assess result sensitivity to these in Figure 9.

F.9 SENSITIVITY ANALYSIS OF THE MODALITY CONTRIBUTION THRESHOLD $\varepsilon$

To analyze the robustness, we selected $\varepsilon$ values from the set $\{0.05, 0.08, 0.1, 0.15, 0.2, 0.3, 0.5\}$, as illustrated in Figure 10. These values are non-uniform because the modality evaluation method is more sensitive to smaller $\varepsilon$, requiring denser sampling for changes, while larger $\varepsilon$ stabilizes (Eq. 10). The results show that a too-small $\varepsilon$ leads to insufficient samples, making it difficult to reflect differences in modality contributions, while a too-large $\varepsilon$ increases noise, affecting quantification accuracy. Regardless of the chosen threshold, our modality evaluation objective effectively balances contributions across modalities.

**Does introducing hyperparameter $\varepsilon$ require extra tuning?** It is imperative to clarify the role and nature of the parameter $\varepsilon$ introduced within our modality contribution assessment framework. This parameter serves a specific, post-hoc analytical purpose and is explicitly decoupled from the model's training, optimization, and inference phases. Consequently, $\varepsilon$ does not constitute a tunable hyperparameter that imposes an additional burden on model development or influences the final predictive performance. Algorithmically, as defined in Equation 10, $\varepsilon$ establishes a tolerance threshold that delineates whether a modality's individual prediction is sufficiently accurate to be classified as a "valid contribution." The selection of a specific value for $\varepsilon$ is supported by a comprehensive sensitivity analysis, detailed in Appendix F.9 and visualized in Figure 10. This analysis demonstrates the robustness of our qualitative conclusions to the precise value of $\varepsilon$. While variations in $\varepsilon$ naturally affect the absolute magnitude and smoothness of the contribution metrics, the core scientific insights remain invariant across a reasonable range (e.g., 0.1 to 0.3). Specifically, two central findings are consistently observed irrespective of the chosen $\varepsilon$:

1. The pronounced predictive dominance of the reaction fingerprint modality on chemically heterogeneous datasets such as ACR.

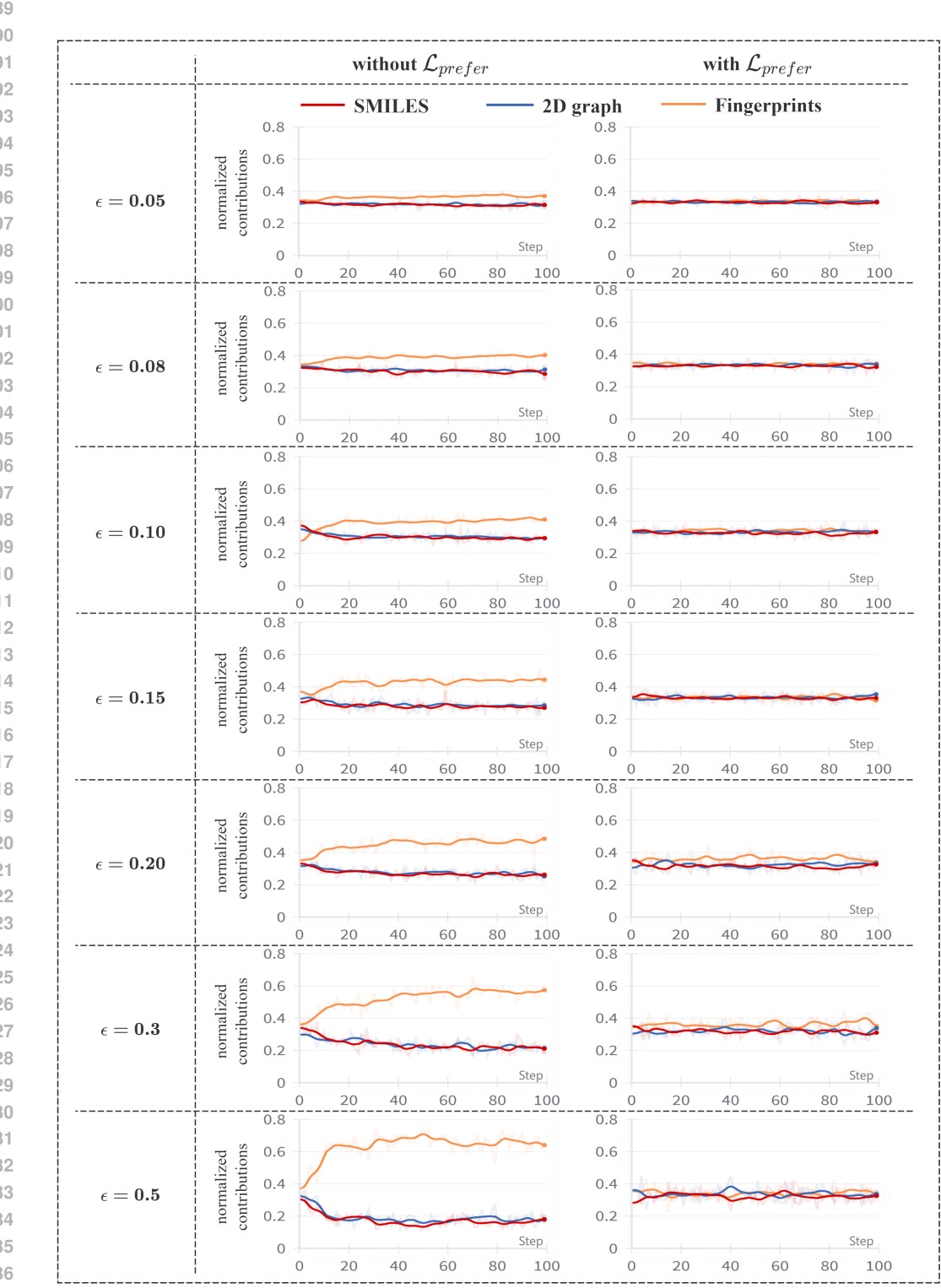

Figure 10: Sensitivity testing of the modality contribution evaluation threshold $\varepsilon$. Contributions are normalized to the range of 0 to 1 using the softmax function.

2. The efficacy of our proposed loss function $\mathcal{L}_{\text{prefer}}$ in fostering a balanced utilization of all modalities, as evidenced in Figure 10.

Therefore, $\varepsilon$ functions solely as a robust probe for interpreting inter-modality dynamics, not as a component of the predictive model itself.

## G  TRAINING EFFICIENCY ANALYSIS

We further analyze the training efficiency of Harmony relative to competitive baselines and within ablation studies, using the challenging ACR dataset as a benchmark. For multimodal models with comparable predictive accuracy, Harmony achieves a favorable balance between performance and efficiency: it is $3.4\times$ faster than YieldGNN and $1.7\times$ faster than UAM, while substantially outperforming ReaMVP, whose speed advantage stems from excessive simplification that compromises predictive power. Quantitatively, Harmony reaches state-of-the-art accuracy ($R^2 = 0.320$) with a per-epoch training cost of $\approx 280s$, compared to $954s$ for YieldGNN and $487s$ for UAM (Table 11).

Table 11: Comparison of Harmony with baseline models on the ACR dataset.

| Method | MAE ↓ | RMSE ↓ | $R^2$ ↑ | Time/Epoch (s) ↓ |
|---|---|---|---|---|
| ReaMVP | 16.02 | 20.51 | 0.201 | $\approx$21 |
| YieldGNN | 15.27 | 19.82 | 0.216 | $\approx$954 |
| UAM | 14.76 | 19.33 | 0.262 | $\approx$487 |
| **Harmony** | **14.72** | **18.88** | **0.320** | $\approx$280 |

Ablation studies provide further insight into the computational profile of each module (Table 12). Removing $\mathcal{L}_{\text{prefer}}$ or $\mathcal{L}_{\text{info}}$ components slightly reduces training time but significantly degrades accuracy, underscoring their necessity. Substituting Mamba with a Transformer of similar size substantially increases runtime, highlighting Mamba's efficiency for fingerprint encoding. Eliminating the 2D graph modality yields the largest time reduction, since our current GNN implementation processes graphs sequentially within a batch; however, this comes at the expense of major accuracy loss, confirming the importance of structural topology. **Taken together, these results demonstrate that Harmony is not only accurate but also computationally efficient, with each architectural choice justified by a favorable trade-off between cost and predictive gain.**

Table 12: Ablation study on Harmony components.

| Model Variant | MAE ↓ | RMSE ↓ | $R^2$ ↑ | Time/Epoch (s) ↓ |
|---|---|---|---|---|
| w/o $\mathcal{L}_{\text{prefer}}$ | 14.77 | 19.45 | 0.278 | $\approx$272 |
| w/o $\mathcal{L}_{\text{info}}$ | 15.22 | 19.47 | 0.276 | $\approx$269 |
| w/o molecular-level fusion | 15.09 | 19.06 | 0.306 | $\approx$265 |
| w/o KAN | 14.76 | 18.91 | 0.317 | $\approx$275 |
| w/o Mamba | 14.98 | 19.06 | 0.306 | $\approx$332 |
| w/o Seq | 14.79 | 19.02 | 0.310 | $\approx$198 |
| w/o Graph | 15.38 | 19.47 | 0.276 | $\approx$108 |
| w/o ECFPs | 16.75 | 20.82 | 0.173 | $\approx$232 |
| **Harmony (full)** | **14.72** | **18.88** | **0.320** | $\approx$280 |

## H  EXPLORATORY EXPERIMENT ON INCORPORATING TEXT EMBEDDING AS AN ADDITIONAL MODALITY

Motivated by the emerging paradigm of integrating scientific AI with large language models (LLMs), which is reshaping computational chemistry, we explored the incorporation of textual descriptions as a fourth modality into the Harmony framework. While traditional molecular representations (e.g., SMILES, 2D graphs, fingerprints) effectively encode static structural information,

the dynamic process of a chemical reaction—encompassing reagents, conditions, and mechanistic context—is more naturally described in natural language. LLMs, trained on vast scientific corpora, can capture such rich semantic knowledge. This experiment investigates whether text embeddings can provide complementary, semantically-rich dynamic information to enhance reaction yield prediction, aligning with the next-generation vision of multimodal chemical foundation models. We extended Harmony by introducing text embeddings at the late-fusion stage: reaction SMILES sequences were converted into descriptive text via the PubChem API, which were then encoded into 1024-dimensional vectors using a frozen, lightweight Qwen3-Embedding-0.6B model. These text embeddings were concatenated with features from the original modalities (SMILES, 2D graphs, fingerprints) and fed into the KAN-based fusion module for prediction. Results on the challenging ACR dataset are compared below:

Table 13: Incorporating Text Embedding as an Additional Modality.

| Model | MAE($\downarrow$) | RMSE($\downarrow$) | R²($\uparrow$) | Trainable Parameters($\downarrow$) | Time per Epoch($\downarrow$) |
|---|---|---|---|---|---|
| Harmony (Original) | 14.72 | 18.88 | 0.320 | 12,591,530 | $\approx 280s$ |
| Harmony + Text Embedding | 14.66 | 18.85 | 0.332 | 13,640,106 | $\approx 1187s$ |

Results on the challenging ACR dataset demonstrate a nuanced trade-off (Table 13): while the incorporation of text embeddings led to a superior predictive performance, as reflected in the improved metrics, it also incurred a substantial increase in computational cost, with the time per epoch rising approximately fourfold. Our proposed Harmony framework, by contrast, achieves a more favorable balance between performance and training efficiency. It delivers highly competitive results while maintaining significantly lower computational demands, making it a more practical and scalable solution for yield prediction tasks. A noteworthy finding emerged from our modality contribution analysis: the text modality contributed nearly on par with molecular fingerprints. This suggests that text embeddings provide a predictive signal of significant value, capturing complementary high-level semantic context about the reaction process, which aligns with the observed performance gain.

In conclusion, this exploratory experiment validates the substantial promise of integrating linguistic understanding via LLMs, as evidenced by the performance improvement with text embeddings. It also highlights the critical challenge of computational efficiency in such large-scale multimodal fusion. The original Harmony framework effectively addresses this challenge by offering a balanced and efficient architecture without relying on heavy-text encoders. Crucially, this design also provides inherent modality extensibility, allowing for the flexible integration of various data types. Future work will focus on developing more parameter-efficient techniques to harness the power of textual semantics while mitigating the computational overhead, striving towards both high-performance and scalable chemical AI models.

## I ANALYSIS OF MODALITY CONTRIBUTION HETEROGENEITY ACROSS DATASETS

The observed heterogeneity in the relative contributions of input modalities across different datasets is a deterministic outcome directly reflecting the intrinsic chemical diversity of each dataset(Figure 4). This variability is principally governed by the diagnostic power of the reaction fingerprint modality in relation to the complexity of the reaction space. A clear dichotomy emerges when comparing datasets of varying chemical diversity:

In datasets characterized by high mechanistic homogeneity and a constrained distribution of reaction templates (e.g., BH, SM), the core bond-forming and -breaking events are highly conserved across entries. Consequently, the reaction fingerprint, which captures these net transformations, provides information that is largely redundant. In this context, the model's predictive performance becomes more reliant on discerning subtle, static differences between substrates, such as steric and electronic effects, which are more effectively encoded by the SMILES and 2D graph modalities. This leads to a more equitable distribution of predictive importance across all three input streams.

Conversely, in chemically heterogeneous datasets encompassing a broad spectrum of reaction classes (e.g., ACR, BH(ELN)), the primary predictive challenge shifts from fine-grained differentiation to the initial, high-level identification of the reaction chemotype. Here, the reaction fingerprint

modality becomes paramount. Its unique ability to abstract the essential dynamic information of the structural transformation—a feature not explicitly present in the static molecular representations—provides a powerful, coarse-grained classification of the reaction. This dynamic information serves as a crucial contextual anchor, which is then refined by the specific, static structural details provided by the SMILES and 2D graph modalities. Therefore, a direct correlation is observed: the predictive indispensability of the reaction fingerprint modality increases proportionally with the mechanistic diversity of the dataset, underscoring its unique role in capturing the salient, dynamic features of chemical transformations in complex scenarios.

## J   ELUCIDATION OF THE HIERARCHICAL FRAMEWORK IN HARMONY

We provide a detailed explication of the "hierarchical" concept integral to the Harmony model's architecture, as discussed in the main manuscript. The central tenet of our hierarchical design is the principle of fusing chemical information at commensurate levels of granularity. This strategy is deliberately employed to maximize the synergistic potential of complementary data modalities.

The hierarchical nature of the Harmony framework is manifested through two complementary and interconnected dimensions: a tiered fusion architecture and a hierarchy of chemical information granularity.

### J.1   TIERED FUSION ARCHITECTURE

Contrary to a monolithic or "flat" fusion approach, our model implements a multi-stage, tiered fusion process designed for controlled and progressive information integration. This structured approach prevents the premature dilution of fine-grained features by higher-level abstractions. The fusion process is organized as follows:

1. **Tier 1: Foundational Feature Extraction.** Each input modality—namely the SMILES string, the 2D molecular graph, and the reaction fingerprint—undergoes an initial, independent feature extraction process using its respective encoder. This stage generates modality-specific latent representations that capture the unique characteristics of each data source.

2. **Tier 2: Molecular-Level Fusion.** The latent representations derived from the SMILES strings and 2D molecular graphs are subjected to an early fusion mechanism. This critical step integrates two distinct but semantically related views of the molecular structure, creating a unified and more robust molecular-level descriptor.

3. **Tier 3: Reaction-Level Integration.** The consolidated molecular-level representation resulting from Tier 2 is subsequently combined with the reaction-level representation (derived from the chemical fingerprint modality) via a late-fusion strategy. This final stage situates the detailed molecular information within the broader context of the chemical transformation.

### J.2   HIERARCHY OF CHEMICAL INFORMATION GRANULARITY

**The tiered fusion architecture is deliberately designed to mirror the inherent hierarchy of abstraction present in chemical data.** Our model explicitly distinguishes between two primary levels of information granularity:

1. Molecular-Level Information: SMILES strings and 2D graphs provide explicit representations of molecular structure, atomic connectivity, and local chemical environments. They operate at the molecular level of abstraction, describing the static state and intrinsic properties of individual chemical entities involved in a reaction.

2. Reaction-Level Information: In contrast, specialized chemical fingerprints (e.g., difference fingerprints) encode holistic information about the chemical transformation itself. They represent a higher level of abstraction—the reaction level—by capturing the net changes, such as bond formation and cleavage, between reactants and products. This modality describes the dynamic process of the reaction rather than the static properties of a single molecule.

The Harmony model's architecture respects this informational hierarchy. It first integrates the complementary molecular-level data streams in Tier 2 to establish a comprehensive understanding of the

molecular components. Subsequently, in Tier 3, this consolidated molecular representation is fused with the overarching reaction-level information. This sequence ensures that a robust characterization of the molecular substrates is achieved before it is contextualized within the scope of the chemical transformation, leading to a more effective and mechanistically aware predictive model.

### J.3 RATIONALE FOR THE ADOPTED HIERARCHICAL FUSION STRATEGY

The rationale for the specific hierarchical fusion order implemented in our model—first integrating molecular-level features (SMILES and 2D graphs) before incorporating the reaction-level fingerprint—is predicated on the principle of fusing information at commensurate levels of abstraction to achieve maximal synergy. This architectural choice was validated through an empirical evaluation of alternative fusion sequences (Table 3), which revealed their inherent limitations:

The optimal configuration involves the initial fusion of SMILES and 2D graph representations. Both modalities operate at the molecular level, yet they provide complementary perspectives: the SMILES string offers a sequential, canonical representation, while the 2D graph explicitly encodes atomic connectivity and topology. Their early integration facilitates the construction of a holistic and contextually enriched molecular descriptor ("$1 + 1 > 2$" synergy), establishing a robust foundation for subsequent processing.

Conversely, alternative fusion sequences were found to be suboptimal. For instance, the direct fusion of a 2D graph (fine-grained, low-level structural data) with a reaction fingerprint (abstract, high-level transformation data) risks a phenomenon of semantic overshadowing. In such a scenario, the high-level features from the fingerprint could prematurely dominate or dilute the nuanced structural details of the graph, leading to an inefficient utilization of information.

Similarly, an initial fusion of SMILES and fingerprints is fundamentally flawed as it omits the indispensable topological information uniquely provided by the 2D graph. This early omission creates an irretrievable information deficit that cannot be fully rectified by a later introduction of the graph modality, thereby constraining the model's capacity to comprehend complex structural relationships.

Therefore, our hierarchical strategy—first consolidating complementary molecular-level information before integrating the overarching reaction-level context—represents the most logically sound and empirically validated architectural choice for effectively modeling chemical reactions.

## K RATIONALE FOR ARCHITECTURAL CHOICES: MAMBA AND KAN

We elucidates the scientific and domain-specific rationale behind the selection of Mamba for fingerprint encoding and Kolmogorov-Arnold Networks (KAN) for the final fusion module. These choices were not merely driven by their novelty but by a careful consideration of how their intrinsic mechanisms align with the fundamental properties of the chemical data they process.

### K.1 MAMBA FOR REACTION FINGERPRINT ENCODING: CAPTURING SPARSE AND SELECTIVE CHEMICAL TRANSFORMATIONS

The task of the fingerprint encoder is to interpret the difference between reactant and product fingerprints. This difference vector, representing the net change in a reaction, has unique characteristics that make traditional sequence models like Transformers or simple MLPs suboptimal.

**Sparsity of Chemical Events** A chemical reaction, even a complex one, typically involves changes at a very small subset of atomic environments. When represented as a high-dimensional fingerprint (e.g., 1024 bits), this translates to a highly sparse signal. Most bits in the reactant and product fingerprints remain unchanged. The crucial information lies in the few "active" bits that flip or change in value, signifying specific bond formations or breakages. An effective model must be adept at identifying and focusing on these sparse, information-rich events while ignoring the vast background of static information.

**Selective Information Compression** The Mamba architecture, built upon Selective State Space Models (SSMs), is exceptionally well-suited for this task. Its core mechanism involves input-

dependent selective gates that control how information flows into and out of its compressed latent state. In the context of reaction fingerprints, this allows Mamba to:

1. **Dynamically filter irrelevant information**: When processing a sequence of fingerprint bits, the selective gates can learn to effectively "ignore" the static bits (where reactant and product values are identical) by minimizing their impact on the evolving state.

2. **Focus on critical transformations**: Conversely, when an "active" bit representing a key structural change is encountered, the gates can "choose" to update the state significantly, effectively capturing the event.

**Superiority over Alternatives**   A Transformer's self-attention mechanism computes a dense $N \times N$ attention matrix, where N is the sequence length (1024). This is computationally expensive and conceptually inefficient for sparse signals, as it forces the model to calculate relationships between all bit pairs, including the vast majority of irrelevant, static ones. This can introduce noise and hinder the model's ability to isolate the true signal. Mamba's linear-time complexity and selective compression offer a more efficient and targeted approach.

A standard Multi-Layer Perceptron (MLP) would treat the fingerprint as a flat, unordered vector, completely disregarding the potential relational information between bit positions. It lacks an intrinsic mechanism to model the sequential or patterned nature of chemical transformations encoded within the fingerprint.

In summary, the choice of Mamba is a principled one, rooted in its inherent ability to perform selective information processing, which directly mirrors the sparse and event-driven nature of chemical transformations encoded in reaction fingerprints.

### K.2   KAN FOR FINAL FUSION: MODELING COMPLEX, HETEROGENEOUS PHYSICOCHEMICAL INTERACTIONS

The final fusion module is designed to predict reaction yield by jointly leveraging two complementary levels of information: molecular-level features, which provide rich, continuous representations of static molecular structures—including topology, connectivity, and local environments—extracted from the fused SMILES and 2D graph encoders; and reaction-level features, which capture abstract, holistic representations of the dynamic chemical transformation itself, derived from the Mamba-based fingerprint encoder.

The mapping between heterogeneous molecular and reaction-level features and reaction yield is highly non-linear, as outcomes are governed by complex physicochemical principles (e.g., steric hindrance, electronic effects, transition state stability). Accurate prediction thus requires models capable of approximating intricate and often unknown functional forms.

To this end, we adopt Kolmogorov–Arnold Networks (KANs) in the fusion stage. Unlike MLPs with fixed activations, KANs parameterize each input–output connection with a learnable spline function, enabling adaptive modeling of heterogeneous feature interactions. For example, the effect of steric features from molecular encoders can be captured with a distinct non-linearity from that used for reaction-type features derived from fingerprints.

By composing such learned functions, KANs approximate complex, high-dimensional mappings more efficiently than MLPs, reducing the need for excessive depth or width. This flexibility makes them particularly suitable for modeling the intricate energy landscapes underlying reaction yields. Moreover, the spline functions on edges can be visualized, offering a potential route toward interpretable feature–yield relationships.

Therefore, selecting KAN for the final fusion stage is a scientifically grounded decision. Its inherent flexibility in learning arbitrary non-linear functions makes it a superior tool for modeling the complex, multi-scale, and heterogeneous physicochemical interactions that govern chemical reaction yield, moving beyond simple model engineering towards a more principled architectural design.

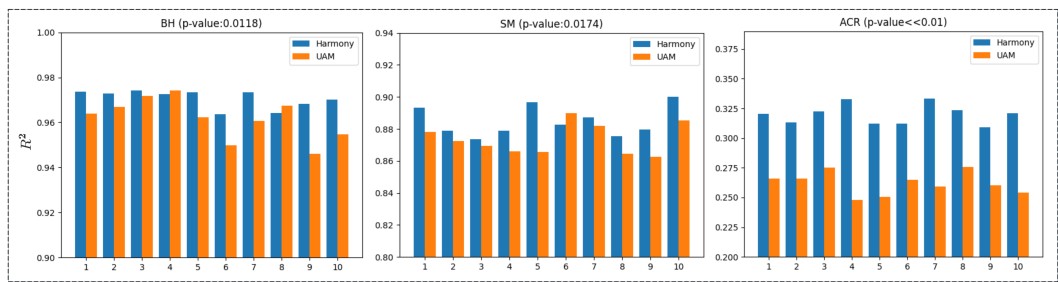

Figure 11: We carried out ten experiments on every benchmark dataset and carried out t-tests on the $R^2$ metric with the help of the Scipy library, and all the p-values were less than 0.05.

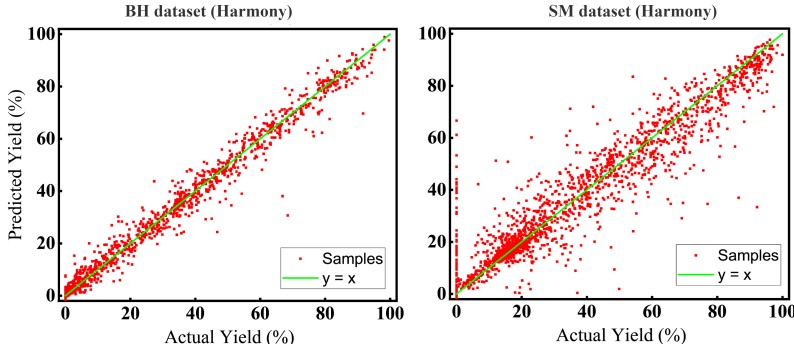

Figure 12: Scatter plots of actual yields versus predicted yields on the test sets of BH and SM datasets for Harmony.

## L    STATISTICAL SIGNIFICANCE TESTING ON BENCHMARK DATASETS

To ensure robust and statistically reliable comparisons, we performed 10 independent data splits on each dataset, training both Harmony and UAM under identical conditions (Table 11). The performance differences between the two models were assessed using paired t-tests. The resulting p-values were 0.012 for BH, 0.017 for SM, and nearly 0 for ACR, all of which are well below the conventional significance threshold of $p < 0.05$. These results provide strong evidence that Harmony achieves statistically significant improvements over UAM across all evaluated datasets. The near-zero p-value for ACR further underscores Harmony's superior performance in this domain.

## M    VISUALIZATION OF PREDICTION RESULTS

We partition datasets according to the dataset partitioning method discussed earlier, and proceed with model training and testing. Figure 12 intuitively demonstrates the predictive performance of our method on the test sets of BH and SM datasets. For the BH dataset, our method achieved an $R^2$ score of 0.979 on the test set, with the scatter plot distribution being very close to the line $y = x$, indicating the best predictive effect. For the SM dataset, our method achieved an $R^2$ score of 0.890 on the test set, with the scatter plot distribution generally close to the line $y = x$. Overall, due to the limited number of reactions and the homogeneity of reaction types in these two datasets, the model is capable of making relatively accurate predictions.

For the ACR dataset, as shown in Figure 13, we compared the predictive results of our method with those of the UAM method, with both being trained on the same training set and tested on the same test set. During the testing phase, our method achieved an $R^2$ of 0.320, while the $R^2$ for UAM was 0.259. From the figure, it can be intuitively seen that the predictions from the Harmony are more concentrated relative to UAM, with fewer deviations from the actual yields, resulting in a better performance.

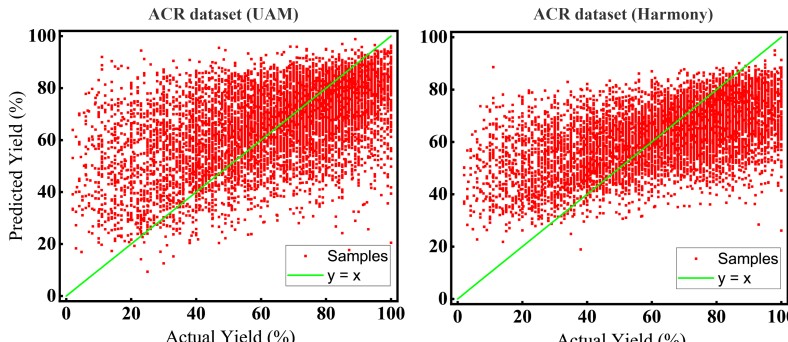

Figure 13: Scatter plots of actual yields versus predicted yields on the test sets of ACR dataset for both UAM and Harmony.

## N ANALYSIS OF THE PREFER-BALANCING OPTIMIZATION OBJECTIVE $\mathcal{L}_{\text{PREFER}}$

### N.1 A SIMPLE ANALYSIS OF $\mathcal{L}_{\text{PREFER}}$ AND ITS EFFECTIVENESS

To mitigate the imbalance phenomenon in the multimodal reaction yield prediction model during the fusion process, we introduce $\mathcal{L}_{\text{prefer}}$. This is a plug-and-play, simple and effective prefer-balancing optimization objective. It maintains linear scalability with the number of modalities, bringing minimal additional overhead. As we mentioned earlier, the goal of $\mathcal{L}_{\text{prefer}}$ is to enhance the ability of each modality $m$ to independently predict results.

To understand it in another way, we can view it as a technique similar to dropout, which randomly drops some modalities and uses the remaining modalities for prediction. Specifically, for $\mathcal{L}_{\text{prefer}}$, it retains only one modality $m$ and discards the data of other modalities. The reason for doing this is that we hope the model can achieve good results regardless of which modalities are discarded, thereby avoiding the model's dependence on certain specific modalities.

To further validate the effectiveness of , we conducted a modality contribution assessment on UAM using our method. Figure 14 shows the comparison of modality contributions before and after using $\mathcal{L}_{\text{prefer}}$ for UAM. By alleviating the modality imbalance with $\mathcal{L}_{\text{prefer}}$, UAM can also achieve performance improvement, as shown in Table 2.

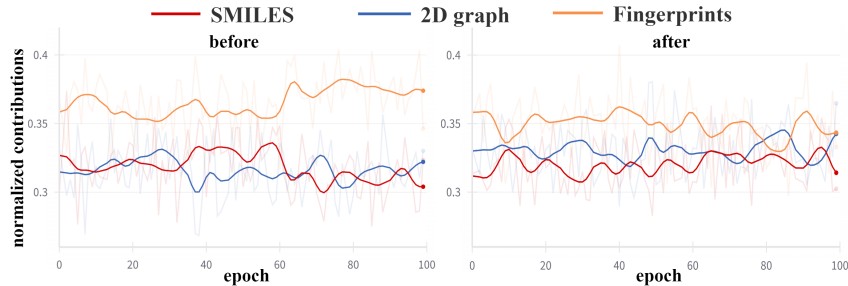

Figure 14: Impact of $\mathcal{L}_{\text{prefer}}$ on the contributions of different modalities within UAM.

### N.2 IS FORCING EACH MODALITY TO PREDICT WELL ALONE LIMITING LEARNING OF MORE SOPHISTICATED COMBINED FEATURES?

**A potential concern is whether the $\mathcal{L}_{\text{prefer}}$ loss function—which forces each modality to become a strong standalone predictor by penalizing its individual prediction errors—theoretically conflicts with the core principle of multimodal learning. This principle relies on leveraging inter-**

**modality complementarity, where a modality that is individually weak as a predictor can still contribute unique information to compensate for the shortcomings of others.**

We concur that the ultimate strength of multimodal fusion resides in synergizing complementary information—where even a modality lacking strong standalone predictive power can deliver crucial, unique signals. The design of $\mathcal{L}_{\text{prefer}}$ is by no means intended to contradict this principle by forcing each modality into a "jack-of-all-trades." Instead, its primary role is to act as a principled regularizer, addressing the well-documented issue of representational overshadowing. Our core argument is that effective synergy can only be constructed on a foundation of high-fidelity, information-rich representations from each individual modality.

We can elaborate on this from two perspectives:

**A Computational and Optimization Perspective** In a standard multimodal architecture without $\mathcal{L}_{\text{prefer}}$, the optimization process is solely driven by the final task loss (e.g., $\mathcal{L}_{\text{mse}}$). If one modality (e.g., reaction fingerprints) offers a much stronger initial gradient signal, the model can quickly converge to a local minimum by predominantly relying on this "easy" source of information. Consequently, the gradients flowing back to the encoders of other, more subtle modalities (e.g., SMILES and 2D graphs) can diminish or vanish. This leads to their encoders remaining under-trained, and their latent representations ($\boldsymbol{h}^{\text{s}}, \boldsymbol{h}^{\text{g}}$) failing to capture the unique physicochemical details they are supposed to encode. The fusion module, therefore, is presented with one high-quality feature vector and several noisy, uninformative ones. At this point, it is impossible to learn any meaningful complementarity because the necessary information from the weaker modalities has already been lost.

$\mathcal{L}_{\text{prefer}}$ directly counteracts this by creating an auxiliary optimization objective that guarantees a persistent and meaningful gradient signal flows back to every modality-specific encoder throughout the training. It compels each encoder to learn a mapping that is, at a minimum, predictive of the target. This does not mean each modality must become a perfect, SOTA predictor on its own. It simply means each must develop an informative latent space. By ensuring a baseline of "individual competence," we provide the final fusion module with a set of rich, diverse, and well-formed feature vectors, creating a fertile ground for it to discover and exploit their complex, synergistic, and potentially asymmetric relationships.

**Chemical Information Perspective** Without $\mathcal{L}_{\text{prefer}}$, the model might discover that reaction fingerprints are excellent for the coarse classification part and achieve a decent $\mathcal{L}_{\text{mse}}$). It might then neglect to train the 2D graph GNN properly, because extracting subtle steric/electronic effects for fine-grained regression is a harder task with a smaller initial payoff. The GNN's output would be noise. The model would never learn the fine-grained regression part.

With $\mathcal{L}_{\text{prefer}}$, we enforce that the 2D graph GNN must also learn to predict the yield. It might not be as good as the fingerprint modality across the board, but it is forced to learn a meaningful representation of molecular structure related to yield. Now, when both the high-level fingerprint embedding and the detailed structural embedding are passed to the final fusion layer, the main $\mathcal{L}_{\text{mse}}$) objective can learn the optimal combination. It is entirely free to learn that the fingerprint feature is most indicative of a reaction being $> 80\%$ yield, but that a specific feature from the 2D graph embedding is what differentiates an 82% yield from a 95% yield.

In summary, $\mathcal{L}_{\text{prefer}}$ does not enforce "homogeneity of function" but rather "universality of information quality." It acts as a foundational regularizer that prevents information loss at the encoding stage, thereby enabling, rather than suppressing, the learning of sophisticated complementary relationships in the subsequent fusion stage. The path from "individual strength" to "collaborative synergy" is predicated on the idea that synergy cannot be constructed from noise. $\mathcal{L}_{\text{prefer}}$ ensures all modalities bring meaningful information to the table, allowing the primary task objective to orchestrate their collaboration most effectively.

### N.3 THE RATIONALE FOR PROMOTING BALANCED MODALITY CONTRIBUTIONS

It is crucial to clarify that our objective is not the rigid enforcement of numerically equal contributions, but rather the implementation of "balance" as a principled regularization strategy to prevent modality dominance. From a chemoinformatic standpoint, each modality provides an orthogonal

yet complementary information stream: SMILES strings capture sequential atomic connectivity, 2D graphs encode explicit molecular topology and bonding, while differential fingerprints encapsulate the high-level dynamics of the chemical transformation. A truly synergistic model must leverage the unique strengths of all three perspectives.

In the absence of explicit regularization, a multimodal model is susceptible to a form of representational "Matthew effect," wherein the optimization process disproportionately relies on the most salient modality (e.g., the reaction fingerprint), particularly in diverse datasets. This heuristic dominance leads to the suppression of subtler, yet critical, static molecular descriptors such as steric and electronic effects, which are primarily encoded in the SMILES and graph representations. Such a scenario fundamentally undermines the rationale for a multimodal architecture, reducing it to a single-modality model with minor perturbations.

Our proposed balancing loss directly counteracts this tendency by enforcing a baseline of predictive autonomy for each modality. This ensures the development of high-fidelity, modality-specific latent spaces prior to their final fusion. By compelling the model to extract meaningful signals from all inputs, we foster enhanced robustness and generalizability, ultimately enabling the emergence of true synergistic effects during the integration stage—a conclusion empirically validated by the model's superior predictive performance.

## O  THE GENERALIZATION CAPABILITY OF HARMONY

### O.1  PERFORMANCE ON THE USPTO (GRAM SCALE) DATASET

In the experimental section, to ensure a fair comparison, we evaluated our method exclusively on the Buchwald-Hartwig (BH), Suzuki-Miyaura (SM), and Amide coupling (ACR) datasets. To further demonstrate scalability, we included additional comparative experiments on the USPTO gram-scale dataset (200k reactions), as presented in Table 14, where our model achieves state-of-the-art (SOTA) performance.

Table 14: Comparative experiments on the USPTO (gram scale) chemical reaction yield prediction dataset.

| Datasets | USPTO (gram scale) | | |
|---|---|---|---|
| Metrics | MAE($\downarrow$) | RMSE($\downarrow$) | $R^2$($\uparrow$) |
| Yield-BERT | - | - | 0.117 |
| DRFP (gradient boost) | - | - | 0.130 |
| UAM | 15.636 | 19.619 | 0.119 |
| Harmony | **15.582** | **19.410** | **0.137** |

### O.2  PERFORMANCE ON THE BH(ELN) DATASET

The BH(ELN) dataset is recognized for its significant challenges, including data sparsity, inherent noise, and a broader, less-controlled distribution of reaction conditions compared to high-throughput experimental (HTE) datasets (e.g., Buchwald-Hartwig dataset and Suzuki-Miyaura dataset). This makes it an ideal benchmark for probing a model's ability to learn robust representations and avoid overfitting to specific data modalities.

To ensure a direct and equitable comparison, our experimental protocol strictly adhered to the setup established by the ReaMVP study. We utilized their officially provided pre-processed data and data splits. The performance of Harmony was benchmarked against the full suite of deep learning and traditional machine learning models reported in the original ReaMVP publication.

The comparative results are summarized in Table 15. Our analysis yields two primary insights:

**State-of-the-Art Performance Among Deep Learning Models**  The results demonstrate that Harmony achieves state-of-the-art performance among all evaluated deep learning models. With an $R^2$ score of 0.263, Harmony surpasses the previous leading model, ReaMVP ($R^2 = 0.212$), by a substantial margin. This performance underscores the efficacy of Harmony's hierarchical fusion archi-

Table 15: Comparative performance of Harmony and baseline models on the BH(ELN) test set. Results for baseline models are cited from the original ReaMVP paper. [a] With RDKit features. [b] Without RDKit features.

| Method | MAE ($\downarrow$) | RMSE ($\downarrow$) | $R^2$ ($\uparrow$) |
|---|---|---|---|
| **RF[a]** | **20.320 (0.769)** | **25.270 (0.937)** | **0.275 (0.040)** |
| RF[b] | 20.560 (0.728) | 25.480 (0.882) | 0.264 (0.032) |
| SVM[a] | 20.900 (0.800) | - | 0.222 (0.057) |
| ContextPred | 22.0 (0.2) | - | 0.177 (0.014) |
| EdgePred | 23.1 (0.2) | - | 0.129 (0.011) |
| AttrMasking | 22.2 (0.2) | - | 0.143 (0.008) |
| W/O pre-training | 22.0 (1.1) | - | 0.132 (0.045) |
| YieldBERT | 22.589 (2.304) | 27.468 (2.005) | 0.143 (0.102) |
| YieldBERT-DA | 21.581 (2.192) | 26.973 (1.981) | 0.171 (0.112) |
| UA-GNN | 20.635 (1.127) | 26.499 (1.027) | 0.203 (0.054) |
| ReaMVP | 20.692 (1.330) | 26.364 (1.289) | 0.212 (0.057) |
| **Harmony (Ours)** | **20.551 (1.112)** | **25.624 (1.164)** | **0.263 (0.059)** |

tecture and, critically, the role of the preference-balancing loss ($\mathcal{L}_{\text{prefer}}$). By preventing the model from over-relying on a single, potentially unreliable modality, a significant risk in noisy datasets like BH(ELN), $\mathcal{L}_{\text{prefer}}$ promotes the learning of more generalizable and robust representations, leading to superior predictive accuracy.

**Narrowing the Performance Gap to Traditional Machine Learning** It is noteworthy that a Random Forest (RF) model leveraging RDKit features (RF[a]) attains a slightly higher $R^2$ of 0.275. This observation aligns with established findings where tree-based models can exhibit strong performance on smaller, structured chemical datasets. However, the crucial insight lies in the performance relative to this strong baseline. While most deep learning approaches exhibit a considerable performance deficit compared to RF[a], Harmony significantly closes this gap. This result provides compelling evidence that the $\mathcal{L}_{\text{prefer}}$ objective functions as an effective regularizer in data-limited and heterogeneous scenarios. It enhances the stability and baseline performance of the deep learning architecture, mitigating the risk of performance collapse due to modality preference or data noise, and thereby enabling it to compete more effectively with traditional, feature-engineered methods.

### O.3 Out-of-Sample Performance on BH and SM Datasets

We have rigorously evaluated Harmony on the out-of-sample (OOS) splits of the Buchwald-Hartwig (BH) and Suzuki-Miyaura (SM) datasets used in the ReaMVP paper. To ensure a fair comparison, we used the pre-split data from the official ReaMVP repository and reported the performance of all models on the $R^2$ metric. We thank the authors of ReaMVP for making their data and code public, which greatly facilitates fair comparisons and rapid progress in the field.

The experimental results provide strong evidence for Harmony's excellent generalization ability (Table 16). Harmony consistently outperformed the previous SOTA model on these highly challenging OOS tasks. In Test 1 of BH, the $R^2$ value of Harmony ($0.884 \pm 0.007$) represents a 4.7% relative improvement over the previously optimal ReaMVP ($0.844 \pm 0.004$). In Test 2 of SM, Harmony ($0.593 \pm 0.016$) increases by approximately $0.059$ compared to ReaMVP ($0.534 \pm 0.018$), with a relative improvement of over 11%, showing a particularly significant advantage.

We believe this superior generalization performance stems from our model's core design:

- **The key role of the $\mathcal{L}_{\text{prefer}}$ loss:** The $\mathcal{L}_{\text{prefer}}$ mechanism forces the model not to over-rely on the "memory" of any single modality (like the specific fingerprint of a common ligand). Instead, it must learn to extract more general chemical principles from SMILES, 2D graphs, and fingerprints together. As a result, even when faced with entirely new ligands or reactants, the model can make accurate predictions based on its fundamental understanding of chemical reactions.

- **The unique advantage of the hierarchical framework:** Our framework distinguishes and integrates molecular-level and reaction-level information, which allows the model to build more comprehensive and robust representations.

In summary, Harmony's excellent performance on the OOS splits is not a coincidence but a direct result of its core mechanisms (especially modality balancing) successfully improving its generalization.

Table 16: Out-of-Sample Performance on BH and SM Datasets

| Dataset | Split type | YieldBERT | YieldBERT-DA | UA-GNN | ReaMVP | **Harmony** |
|---------|-----------|-----------|--------------|--------|--------|-------------|
| BH | Test 1 | $0.824 \pm 0.010$ | $0.811 \pm 0.047$ | $0.744 \pm 0.042$ | $0.844 \pm 0.004$ | $\mathbf{0.884 \pm 0.007}$ |
| | Test 2 | $0.829 \pm 0.037$ | $0.866 \pm 0.020$ | $0.876 \pm 0.026$ | $0.896 \pm 0.004$ | $\mathbf{0.913 \pm 0.004}$ |
| | Test 3 | $0.741 \pm 0.030$ | $0.585 \pm 0.067$ | $0.717 \pm 0.024$ | $0.792 \pm 0.025$ | $\mathbf{0.801 \pm 0.029}$ |
| | Test 4 | $0.444 \pm 0.077$ | $0.157 \pm 0.034$ | $0.496 \pm 0.031$ | $0.693 \pm 0.038$ | $\mathbf{0.717 \pm 0.035}$ |
| | Plate 1 | $0.752 \pm 0.012$ | $0.789 \pm 0.013$ | $0.730 \pm 0.037$ | $0.785 \pm 0.011$ | $\mathbf{0.795 \pm 0.010}$ |
| | Plate 2 | $0.181 \pm 0.011$ | $0.334 \pm 0.023$ | $0.202 \pm 0.121$ | $0.349 \pm 0.129$ | $\mathbf{0.421 \pm 0.033}$ |
| | Plate 3 | $0.718 \pm 0.014$ | $0.669 \pm 0.056$ | $0.787 \pm 0.023$ | $0.779 \pm 0.017$ | $\mathbf{0.794 \pm 0.015}$ |
| | Plate 2 new | $0.508 \pm 0.010$ | $0.566 \pm 0.014$ | $0.451 \pm 0.083$ | $0.689 \pm 0.026$ | $\mathbf{0.697 \pm 0.026}$ |
| SM | Test1 | $0.306 \pm 0.005$ | $0.307 \pm 0.012$ | $0.462 \pm 0.400$ | $0.574 \pm 0.033$ | $\mathbf{0.596 \pm 0.023}$ |
| | Test2 | $0.469 \pm 0.021$ | $0.467 \pm 0.014$ | $0.420 \pm 0.022$ | $0.534 \pm 0.018$ | $\mathbf{0.593 \pm 0.016}$ |
| | Test3 | $0.357 \pm 0.024$ | $0.395 \pm 0.025$ | $0.417 \pm 0.021$ | $0.468 \pm 0.017$ | $\mathbf{0.474 \pm 0.027}$ |
| | Test4 | $0.239 \pm 0.008$ | $0.229 \pm 0.010$ | $0.299 \pm 0.017$ | $0.323 \pm 0.043$ | $\mathbf{0.342 \pm 0.036}$ |

## P MORE DETAILS ABOUT ENCODERS

### P.1 SMILES ENCODER

The Transformer-based architecture is a common way to encode character sequences(Li & Fourches, 2021; Soares et al., 2024; Zheng & Tomiura, 2024). Due to limited data in yield prediction datasets, the SMILES encoder may prioritize learning the format over underlying chemical principles of SMILES. Fig7 in SciInstruct(Zhang et al., 2024) shows the model understands scientific principles well only when data volume exceeds a threshold (e.g. 120k). Meanwhile, the graph encoder effectively extracts atomic and bond features from 2D graphs without capturing irrelevant information.

Although special tokens such as '.' and '>' were not present in the pre-training dataset, the ChemBERTa-2 model is equipped to handle them. It consists of a tokenizer and a Transformer-based encoder. Notably, the tokenizer's vocabulary already encompasses symbols like '.' and '>'. These symbols are converted into tokens in the same way as other characters. During the training process, to further enhance the model's ability to process these special symbols, we unfroze the parameters of the encoder's last head layer. This adjustment allows the model to learn how to effectively process these special symbols.

It is crucial to clarify that this approach does not introduce data leakage, as ChemBERTa-2 is pre-trained exclusively on unsupervised tasks, such as masked language modeling, without any exposure to the reaction yield labels used in our downstream task. Its role is strictly limited to feature extraction, not yield prediction.

To validate our design choice and quantify the impact of the pre-trained encoder, we conducted an ablation study comparing three distinct configurations. The results are summarized in Table [Your Table Number]. The configurations are:

1. Training ChemBERTa-2 from scratch: The entire ChemBERTa-2 architecture is trained end-to-end with our model, without leveraging its pre-trained weights.

2. Custom lightweight encoder: We replaced ChemBERTa-2 with a custom, smaller Transformer-based encoder that is trained from scratch.

3. Our proposed method (Harmony): We use the pre-trained ChemBERTa-2 as a frozen feature extractor, fine-tuning only the final projection layer.

Table 17: Performance and parameter efficiency comparison of different SMILES encoder strategies. Our proposed method (Harmony), which leverages a frozen pre-trained ChemBERTa-2, achieves the best results with the fewest trainable parameters.

| Method | MAE ($\downarrow$) | RMSE ($\downarrow$) | $R^2$ ($\uparrow$) | Trainable Parameters ($\downarrow$) |
|---|---|---|---|---|
| (1) Training ChemBERTa-2 from scratch | 14.84 | 19.31 | 0.293 | 56,104,874 |
| (2) Custom lightweight SMILES encoder | 14.77 | 18.94 | 0.316 | 18,958,765 |
| (3) Our Method (Harmony) | 14.72 | 18.88 | 0.320 | 12,591,530 |

As shown in Table 17, our approach (3) achieves the best performance across all metrics while requiring the fewest trainable parameters (12.6M). The custom lightweight encoder (2) provides comparable results but with a 50% increase in parameters. Notably, training the full ChemBERTa-2 model from scratch (1) significantly degrades performance and results in a parameter count over four times larger than our method, substantially increasing the risk of overfitting.

These findings strongly suggest that the performance improvement stems from the powerful and generalizable feature extraction capabilities of the pre-trained ChemBERTa-2 model, rather than any form of data leakage. Our proposed design effectively harnesses this pre-trained knowledge, leading to better predictive accuracy, enhanced parameter efficiency, and faster convergence. This validates the effectiveness of our Harmony architecture.

**Why is the SMILES modality, despite extensive pre-training on structural data, the least important?** An analysis of the ablation studies revealed that the SMILES modality, despite extensive pre-training on large-scale chemical structure corpora, exhibited a comparatively subordinate contribution to the model's overall predictive accuracy. This observation can be rationalized by considering the fundamental physicochemical determinants that govern chemical reaction yields. The quantitative prediction of reaction outcomes is intrinsically sensitive to subtle steric and electronic effects, as well as the net structural transformations occurring between reactants and products. Within our multimodal framework, the 2D molecular graph explicitly encodes atomic connectivity and topology, while the reaction fingerprint directly quantifies the aggregate bond changes inherent to the transformation. In contrast, the SMILES modality, as a one-dimensional linearized notation, possesses intrinsic limitations in explicitly representing the three-dimensional spatial arrangements and electron density distributions that underpin these critical physicochemical phenomena. Although pre-training endows the SMILES representation with a rich, latent understanding of chemical principles, its inherent topological nature constrains its ability to convey the geometric and electronic information as effectively as the other modalities. This representational gap logically explains its diminished relative importance and strongly motivates the future integration of 3D conformational data as a promising avenue to enhance predictive power by more directly capturing these essential drivers of chemical reactivity.

## P.2 2D GRAPH ENCODER

Following the established practice from (Kwon et al., 2022) and (Chen et al., 2024), we employ a commonly adopted graph neural network (GNN) encoder for 2D molecular graph feature extraction. This encoder has been widely validated for its effectiveness in capturing both local atomic environments and global molecular structures. This choice ensures a fair comparison with existing methods while maintaining a strong representational capacity for downstream tasks.

## P.3 FINGERPRINTS ENCODER

For fingerprints, a dynamic time-varying parameterized feature selection mechanism based on the Mamba state space model dynamically focuses via gating units on features with significant absolute fingerprint changes and features with relative proportion changes. This dynamic mechanism outperforms MLP/attention-based static processing.

## Q  LIMITATIONS AND FUTURE WORK

Although Harmony effectively improved the performance of reaction yield prediction through a hierarchical fusion framework and balanced modality fusion, there are still further optimization points, which will be our next direction for improvement.

Firstly, most reaction yield prediction models, including Harmony, are currently unable to handle reaction conditions. For the same chemical reaction, different reaction temperatures, reaction times, solvents, reagents, and other conditions can all affect the actual yield of the reaction. Integrating reaction conditions into the yield prediction can further narrow down the search space and provide more comprehensive information for yield prediction. Our next step will involve considering how to incorporate reaction condition information into the model to achieve more practical and accurate yield predictions.

Additionally, we have designed a prefer-balancing optimization objective to balance the contributions of different modalities during the fusion process. Although this design is effective, there is still room for improvement, such as starting directly from gradients, which will be one of our next research directions.

## R  IMPACT STATEMENT

This paper presents work aimed at advancing the field of yield prediction for chemical reactions. Enhancing the accuracy and efficiency of yield prediction can bring substantial social benefits to various fields.

For chemical companies, accurate yield predictions optimize production by enabling precise raw material procurement based on forecasted yields. In large-scale pharmaceutical manufacturing, this not only ensures enough product to meet market demand but also cuts down on raw material waste, reduces production costs, and boosts economic efficiency.

Furthermore, our method enhances AI-assisted synthesis prediction, boosting research and development (R&D) efficiency. By facilitating faster screening of potential reaction pathways through yield prediction, it shortens the research and development cycle and reduces costs in developing new chemical substances.

On a macro scale, accurately predicting chemical reaction yields aids in the efficient distribution of chemical resources society-wide. For instance, precise yield forecasts for rare metals or hazardous chemicals can prevent overuse, ensuring their sustainable utilization and minimizing environmental risks.

Generally speaking, this work broadly offers potential social benefits, including advancements in chemical production, scientific progress, and sustainable resource management.

## S  PSEUDOCODE FOR MODALITY CONTRIBUTION EVALUATION ALGORITHM

We provide Algorithm 1, which presents the pseudocode for the process of evaluating contributions of each modality.

Taking the computation of the contribution of modality $m_j$ as an example, the overall algorithm can be divided into three steps:

1. The first step is model forward propagation, where predictions are made using all modalities $\mathcal{M}$ and the modality subset $\mathcal{C}$, corresponding to Equation equation 8. Here, the modality subset $\mathcal{C}$ consists of the remaining modalities in $\mathcal{M}$ after removing $m_j$, that is, $\mathcal{C} = \mathcal{M} \setminus \{m_j\}$.

2. The second step involves calculating the contributions of both the full set of modalities $\mathcal{M}$ and the modality subset $\mathcal{C}$ to the prediction outcome, using Equation equation 10 to compute $\mathcal{B}(\mathcal{M})$ and $\mathcal{B}(\mathcal{M} \setminus \{m_j\})$.

3. The third step uses Equation equation 13 to calculate the contribution of the single modality $m_j$, $\beta(m_j)$.

**Algorithm 1** Calculating Contributions of Each Modality

---

**Input:** $\mathcal{M} = \{m | m \in \{s, g, f\}\}$: The set of all modalities, where s, g, f correspond to SMILES, 2D graphs, and fingerprints, respectively

$\{(\boldsymbol{x}_i, y_i)\}_{i=1}^N$: All samples in the dataset

$\mathcal{F}(\cdot)$: Late fusion module

$\Phi^m(\cdot)$: Feature extractor for modality $m$

$\varepsilon$: Threshold for contribution calculation

$\delta$: A small constant introduced to prevent numeric overflow

        ▷ stage 1. Predict yield using both the full modality set $\mathcal{M}$ and modality subset $\mathcal{C}$.

1: **for** $i$ from 1 to $N$ **do**

2:     $\hat{y}_i^{\mathcal{M}} = \mathcal{F}(\Phi^{m_1}(\boldsymbol{x}_i^{m_1}) \oplus \Phi^{m_2}(\boldsymbol{x}_i^{m_2}) \oplus \cdots \oplus \Phi^{m_n}(\boldsymbol{x}_i^{m_n}))$        ▷ Predict yield using the complete modality set $\mathcal{M}$, where $\oplus$ denotes the concatenate operator.

3:     **for** $j$ from 1 to $n$ **do**

4:        $\hat{y}_i^{\mathcal{M} \setminus \{m_j\}} = \mathcal{F}(\Phi^{m_1}(\boldsymbol{x}_i^{m_1}) \oplus \Phi^{m_2}(\boldsymbol{x}_i^{m_2}) \oplus \cdots \oplus \Phi^{m_{j-1}}(\boldsymbol{x}_i^{m_{j-1}}) \oplus \mathbf{0} \oplus \Phi^{m_{j+1}}(\boldsymbol{x}_i^{m_{j+1}}) \oplus \cdots \oplus \Phi^{m_n}(\boldsymbol{x}_i^{m_n})))$        ▷ Predict yield using the complete modality set $\mathcal{M}$, excluding modality $m_j$.

5:     **end for**

6: **end for**

        ▷ stage 2. Calculate contributions of both the full modality set $\mathcal{M}$ and modality subset $\mathcal{C}$.

7: **for** $i$ from 1 to $N$ **do**

8:     **if** $|\hat{y}_i^{\mathcal{M}} - y_i| < \varepsilon$ **then**

9:        $\mathcal{B}(\mathcal{M}) = \frac{|\mathcal{M}|}{N} \cdot \sum_{i=1}^N \min\left(1, 2 \cdot \log \frac{\varepsilon}{|\hat{y}_i^{\mathcal{M}} - y_i| + \delta}\right)$        ▷ Calculate contribution of the complete modality set $\mathcal{M}$.

10:     **else**

11:        $\mathcal{B}(\mathcal{M}) = 0$

12:     **end if**

13:     **for** $j$ from 1 to $n$ **do**

14:        **if** $|\hat{y}_i^{\mathcal{M} \setminus \{m_j\}} - y_i| < \varepsilon$ **then**

15:           $\mathcal{B}(\mathcal{M} \setminus \{m_j\}) = \frac{|\mathcal{M}|}{N} \cdot \sum_{i=1}^N \min\left(1, 2 \cdot \log \frac{\varepsilon}{|\hat{y}_i^{\mathcal{M} \setminus \{m_j\}} - y_i| + \delta}\right)$        ▷ Calculate contribution of the complete modality set $\mathcal{M}$, excluding modality $m_j$.

16:        **else**

17:           $\mathcal{B}(\mathcal{M} \setminus \{m_j\}) = 0$

18:        **end if**

19:     **end for**

20: **end for**

        ▷ stage 3. Calculate single modality contributions using the contributions of full modality set $\mathcal{M}$ and modality subset $\mathcal{C}$.

21: **for** $j$ from 1 to $n$ **do**

22:     $\beta(m_j) = \mathcal{B}(\mathcal{M}) - \mathcal{B}(\mathcal{M} \setminus \{m_j\})$     ▷ Calculate the contribution of each single modality $m_j$ within the complete modality set $\mathcal{M}$.

23: **end for**

**Output:** Contributions of each modality in $\mathcal{M}$.

---

