# OpenReview forum: "Finding Harmony in Chemical Data: Hierarchical and Balanced multimodal Fusion for Reaction Yield Prediction"
_ICLR.cc/2026/Conference — ICLR 2026 Conference Withdrawn Submission_

### Official Review · Reviewer_cBXr · 2025-10-23

**Soundness:** 2
**Presentation:** 3
**Contribution:** 2
**Rating:** 2
**Confidence:** 3

**Summary:**

Challenges. Early models can only utilize a single modality, which can not capture the full features from structural, topological, and electrical aspects. 1) "Flat fusion" approaches lead to feature interference, so that important signals are obscured by noise. 2) In prior work UAM, some modalities were not modeled very well, showing a foundational flaw in multimodal chemical modeling.

Novelties:
1. The proposed Harmony leverages the hierarchical and balanced fusion framework, integrating molecular-level and fingerprint-level features to prevent modality collapse.
2. A module can balance modality contribution to ensure real engagement of each chemical feature.

Results:
Top performance on three datasets and strong generalization on out-of-sample data. 22% improvement on the claimed "most challenging dataset".

**Strengths:**

1. This paper is a good attempt at using multimodal methods in reaction yield prediction. The experiments show relatively small but consistent improvement on the benchmark datasets.
2. The motivation from "Flat fusion" is typical, and one of the novelties of this work is trying to figure out modality contribution and pursue balance.
3. The paper is well organized and written. The logic and the structure is clear and easy to follow.

**Weaknesses:**

1. Figure 2 shows that contrastive learning is established between the fusion feature of SMILES and the graph with fingerprint features; however, the unlabeled equation in line 344 shows it is between the graph feature and the fingerprint feature, which is not consistent. If it is for only graphs and fingerprints, the claim of hierarchy will not be correct.
2. The novelty may not be strong enough. This work mainly puts existing modules (such as Mamba layer, molecular encoders, MLP) together to make a multimodal pipeline.  The pipeline framework may be simple to use concatenation and MLP blocks. There may not be enough improvement beyond the limited 'flat fusion'. From the code, I see that contrastive learning is only for the graph feature with fingerprint; there is no SMILES feature engaged.
3. The hierarchical information flow is only serving among three modalities -- SMILES, graph, and fingerprint. The first level is between SMILES and graph, the second is between graph and fingerprint. In each level, the features are fused by "Concat" and modeled by MLP (one mamba layer added for fingerprint) instead of choosing the most suitable modeling module and best fusion approach for modalities.
4. The selection rationale of each module has not been explained very well. For instance, why add a Mamba layer for fingerprint data, why use contrastive learning only for graph and fingerprint features, or the reasoning for choosing each molecular encoder?
5. The scope of this work may be relatively narrow. This framework can only be used in reaction yield prediction due to its fixed framework design, limiting the contribution and potential to generalize to broader multimodal fields.
6. The main results only have 3 datasets, which may not be enough to prove the performance of this model.

**Questions:**

1. Related works have used SMILES, 2D, and 3D conformers for the integration and claimed strong performance. Why does this work only consider SMILES and 2D structures? Can 3D conformers further enhance the performance?
2. For the integration of SMILES and 2D, the work uses two different encoders for each modality. However, the multimodal molecular modeling field has been explored deeply -- why not use a multimodal molecular encoder that can integrate SMILES and gthe raph itself, which may gain more performance?
3. Following the above question, this work uses a relatively old ChemBERTa-2 (2022) model as its SMILES encoder. Why not try a later one, such as TranFoxMol (2023), MLM-FG (2024), MuMo (2025), with stronger performance?
4. Did you specify which 2D graph encoder you are using as Enc_g?
5. The output from Molecular-level Fusion has already integrated the SMILES information, why concatenated with SMILES again? Will this enhance the contribution of the SMILES modality a lot?
6. In the main model, this work mainly uses MLP as a modeling module for the concatenated features/modalities. Did you try a newer modeling backbone, such as transformers or Mamba-ssm?

---

### Official Review · Reviewer_xuiP · 2025-10-29

**Soundness:** 3
**Presentation:** 3
**Contribution:** 2
**Rating:** 6
**Confidence:** 4

**Summary:**

The paper addresses reaction yield prediction by integrating multiple chemical modalities (SMILES, molecular graphs, and reaction fingerprints) while mitigating modality dominance during fusion. It proposes Harmony, a hierarchical fusion architecture. Three features from different modality encoders are hierarchically mixed, and the final prediction is yielded with a KAN head with Gaussian output. On top of training that combines MSE/uncertainty loss/contrastive alignment loss, the authors introduced a preference-balancing objective that supervises per-modality heads to quantify and balance contributions. Experiments on different reaction experiment datasets compare against unimodal and multimodal baselines show lower MAE/RMSE and higher $R^2$, with ablations supporting the hierarchy and balancing components.

**Strengths:**

- To maximize the benefits of multimodality, the paper introduces a preference-balancing loss that trains modality-specific predictors in isolation, thereby encouraging balanced contributions when fused.

- The proposed method consistently achieves superior quantitative performance across datasets and evaluation metrics compared with prior baselines.

- Through very extensive ablation studies and systematic comparisons, the paper provides clear justification for the major design choices made during model construction.

**Weaknesses:**

- Considering that many concepts(e.g., the use of multimodality across SMILES/graphs/fingerprints, contrastive loss alignment, and reparameterization-based range prediction) have already been introduced by prior baseline UAM, certain components for performance improvement (e.g., the adoption of the Mamba and KAN architectures) may fall within the realm of engineering choices rather than constituting substantial methodological advances.

**Questions:**

.

---

### Official Review · Reviewer_8XR5 · 2025-11-01

**Soundness:** 3
**Presentation:** 2
**Contribution:** 3
**Rating:** 6
**Confidence:** 3

**Summary:**

This paper proposes Harmony, a hierarchical multimodal fusion model for reaction yield prediction. The method uses SMILES strings, 2D graphs, and molecular fingerprints to encode moleular and reaction-level embeddings, using contrastive learning and a preference-balancing loss to train their method. Experiments show promising performance compared to competing methods.

**Strengths:**

- The method consistently outperforms baselines
- Their combination of specific losses make sense, especially $\mathcal{L}_{info}, \mathcal{L}_{prefer}$.

**Weaknesses:**

- Some architecture decisions are underexplained, i.e. the use of Mamba for fingerprint embeddings, the use of KANs in the MLPBlock. While Appendix K.2 claims to justify the use of a KAN, it is not convincing given that traditional deep learning components ahve been shown to model "heterogeneous feature interactions." Further, their Mamba justification ignores the fact that fingerprint sequences are fixed one-hot descriptors, not a long, variable-length sequences. Using a sequence-based model for the fingerprints seems overkill, and it's unclear why a transformer/Mamba is considered for this. Exact architecture of the 2D graph encoder is also unspecified.
- The “causal/counterfactual” contribution estimator feels ad hoc and slightly trivial; its $\xi$ definition literally comes from massaging an error term into (0,1]). There are no results or theoretical analyses that the resulting scores line up with standard Shapley-style methods.
- The paper's writing feels salesy and tends to overclaim. For example, they claim their rationale for KAN is a "scientifically grounded decision" but only high-level discussion and a single ablation are provided as evidence.

**Questions:**

- What is the performance of the method for the "w/o $\mathcal{L}_{prefer}$" ablation?
- Why is the InfoNCE contrastive loss only defined for 2D graphs and fingerprints?
- Are there ablations for the uncertainty loss?
- Have the authors also tested baselines using their other loss functions as well, e.g. the uncertainty loss and InfoNCE loss? There are a lot of components to this method, and it's unclear whether the biggest performance improvements are because of methods proposed by the paper versus the accumulation of general engineering tricks.

---

### Official Review · Reviewer_hceG · 2025-11-01

**Soundness:** 2
**Presentation:** 2
**Contribution:** 2
**Rating:** 2
**Confidence:** 3

**Summary:**

The authors present a HARMONY framework to predict reaction yield. They specifically leverage hierarchical NN architectures multimodal information (i.e., SMILES strings, graphs and fingerprints) as well as balance different NNs in charge of their unimodal parts.
The performance evaluation of HARMONY was performed with three datasets (Buchwald-Hartwig, Suzuki-Miyaura and ACR) in terms of MAE, RMSE and R^2 scores, and the advantages of HARMONY are discussed from various aspects including comparisons against other existing approaches, ablation study, and trainable parameter size.

Yield reaction is an important problem in the AI and chemo-informatics research communities.
However, while HARMONY might be an important framework, I have a difficulty in understanding the research motivation on balancing the model.
Each model has features that are also commonly/easily identified by the others, while it has its own strengths and weaknesses.
Without seeing a combination of other types of the unimodal models under the HARMONY framework, the fact that model imbalance was an issue might happen to be the set of the models they happened to select.

Their analysis essentially shows only numbers of MAE, MSE and R2 scores,
and the analysis is shallow in terms of understanding the results from the trends of the molecular structures of reactants and products and the reaction patterns (e.g., structural changes) in the datasets.
For example, if complicated rings exist, atom-atom distances of a Transformer-based approach receiving SMILES string as a weakness might be misidentified, leading to poor performance. Analogously, an MPNN might fail to handle global relations of important substructures if their distances are long, etc.
Although the molecules and their reaction patterns are important factor in determining the performance, it is unclear what kind of weakness each unimodal had in each dataset and how frequently HARMONY resolved these weaknesses.

**Strengths:**

Attempt to improve the performance of reaction prediction

**Weaknesses:**

Unclear motivation to the research question on whether balanced multimodal fusion is a generic principle or not due to a combination of only specific models

Lack of analysis of the pros and cons of the unimodal models and HARMONY from a viewpoint of actual molecular structures and patterns of the reactions

**Questions:**

How would the empirical performance fare if a different set of models is combined to construct HARMONY? Could you provide numbers?

I would like to see a deeper understanding relations between (1) the datasets on the molecular structures of the products and reactants, and reaction patterns and (2) the unimodal/HARMONY models (e.g., weakness of a unimodal model and how and why HARMONY resolved).  Would it be possible to provide evidence showing the trends of all the models for deeper understanding to what happens?

---

### Note · Authors · 2025-12-03

I have read and agree with the venue's withdrawal policy on behalf of myself and my co-authors.